

# Reservoir Computing in R: a Tutorial for Using reservoirnet to Predict Complex Time-Series

ISSN 2824-7795

Thomas Ferté    Inserm Bordeaux Population Health Research Center UMR 1219, Inria BSO, team SISTM, F-33000 Bordeaux, France, Inserm, Inria

Bordeaux Hospital University Center, Pôle de santé publique, Service d'information médicale, F-33000 Bordeaux, France, CHU de Bordeaux

Kalidou Ba    Inserm Bordeaux Population Health Research Center UMR 1219, Inria BSO, team SISTM, F-33000 Bordeaux, France, Inserm, Inria

Dan Dutartre    Inria BSO, Inria

Pierrick Legrand    Inria BSO, F-33000 Bordeaux, France, Inria

IMB, Institut de Mathématiques de Bordeaux, UMR CNRS 5251, IMB

Vianney Jouhet    Inserm Bordeaux Population Health Research Center UMR 1219, team AHeaD, F-33000 Bordeaux, Inserm

Bordeaux Hospital University Center, Pôle de santé publique, Service d'information médicale, F-33000 Bordeaux, France, CHU de Bordeaux

Rodolphe Thiébaut    Inserm Bordeaux Population Health Research Center UMR 1219, Inria BSO, team SISTM, F-33000 Bordeaux, France, Inserm, Inria

Bordeaux Hospital University Center, Pôle de santé publique, Service d'information médicale, F-33000 Bordeaux, France, CHU de Bordeaux

Xavier Hinaut    Inria BSO, F-33000 Bordeaux, France, Inria

Univ. Bordeaux, CNRS, IMN, UMR 5293, Bordeaux, France, IMN

LaBRI, Univ. Bordeaux, Bordeaux INP, CNRS UMR 5800., LaBRI

Boris Hejblum[1]    Inserm Bordeaux Population Health Research Center UMR 1219, Inria BSO, team SISTM, F-33000 Bordeaux, France, Inserm, Inria

Date published: 2025-04-21    Last modified: 2025-04-21

## Abstract

Reservoir Computing (RC) is a machine learning method based on neural networks that efficiently process information generated by dynamical systems. It has been successful in solving various tasks including time series forecasting, language processing or voice processing. RC is implemented in `Python` and `Julia` but not in `R`. This article introduces `reservoirnet`, an `R` package providing access to the `Python` API `ReservoirPy`, allowing `R` users to harness the power of reservoir computing. This article provides an introduction to the fundamentals of RC and showcases its real-world applicability through three distinct sections. First, we cover the foundational concepts of RC, setting the stage for understanding its capabilities. Next, we delve into the practical usage of `reservoirnet` through two illustrative examples. These examples demonstrate how it can be applied to real-world problems, specifically, regression of COVID-19 hospitalizations and classification of Japanese vowels. Finally, we present a comprehensive analysis of a real-world application of `reservoirnet`, where it was used to forecast COVID-19 hospitalizations at Bordeaux University Hospital using public data and electronic health records.

*Keywords:* Reservoir Computing, Covid-19, Electronic Health Records, Time series

---

[1]Corresponding author: boris.hejblum@u-bordeaux.fr

# Contents

# 1 Introduction

Reservoir Computing (RC) is a prominent machine learning method, proposed by Jaeger (2001), Maass, Natschläger, and Markram (2002) and Lukoševičius and Jaeger (2009) that has gained significant attention in recent years for its ability to efficiently process information generated by dynamical systems. This innovative approach leverages the dynamics of a high-dimensional "reservoir" (we define it below) to perform complex computations and solve various tasks based on the response of this dynamical system to input signals. RC has demonstrated its efficacy in tackling various challenges, encompassing pattern classification and time series forecasting in applications ranging from electrocardiogram analysis to bird calls (Trouvain and Hinaut 2021), language processing (Hinaut and Dominey 2013), power plants, internet traffic, stock prices, and beyond (Lukoševičius

and Jaeger 2009; Tanaka et al. 2019).

Originally, the RC paradigm was implemented in artificial firing-rate neurons ("Echo State Networks", Jaeger (2001)) and spiking neurons ("Liquid State Machine", Maass, Natschläger, and Markram (2002)) as a recurrent neural network (RNN) where the internal recurrent connections, denoted as the reservoir, are randomly generated and only the output layer (named "read-out") is trained. The reservoir projects temporal input signals onto a high-dimensional feature space, facilitating the learning of non-linear and temporal interactions. Thus, this recurrent layer contains high-dimensional non-linear recombination of the inputs and past states: it is a "reservoir of computations" from which useful information can be linearly extracted (or "read-out") to provide the desired outputs. This offers the advantage of decreasing the computing time compared to conventional RNNs while consistently maintaining performance (Vlachas et al. 2020). Besides, this RC paradigm fostered increasing interest thanks to its ability to be implemented on classical computers, as the hidden recurrent layer can be kept untrained. A wide range of physical media can be also used to replace it and Tanaka et al. (2019) recently reviewed this prolific field: from FPGA hardware (Penkovsky, Larger, and Brunner 2018), to spin waves using magnetic properties (Nakane, Tanaka, and Hirose 2018), skrymions (Prychynenko et al. 2018) or optical implementations (Rafayelyan et al. 2020). This provides interesting and potentially more efficient alternative to traditional machine learning computing and might play an important role in the coming years (Yan et al. 2024).

RC leverages various hyperparameters to introduce prior knowledge about the relationship between input variables and output targets. But because the connections within the reservoir are randomly initialized, the same set of hyperparameters may exhibit diverse behaviors across different instances of the reservoir connections. This unpredictability makes it challenging to anticipate the performance of a particular hyperparameter setting, as identical settings may produce varying outcomes when applied to distinct instances of the reservoir. Moreover, selecting the most suitable hyperparameters often requires researchers to experiment with multiple combinations on a training dataset and evaluate their performance on a separate test set[2]. Although this approach can be resource-intensive and time-consuming, it is a compromise that is acceptable considering the rapid simulation capabilities offered by RC. Furthermore, there is a current absence of implementation in R, rendering the method challenging for users unfamiliar with Python (Trouvain and Hinaut 2022) or Julia (Martinuzzi et al. 2022).

Here, we offer comprehensive guidance to assist new users in maximizing the benefits of RC. Initially, a broad introduction to reservoir computing is presented in Section 2, followed in Section 3 by a tutorial on its application using reservoirnet, an R package built upon the ReservoirPy Python module (Trouvain, Rougier, and Hinaut 2022; Trouvain and Hinaut 2022; Trouvain et al. 2020). Section 3 then introduces the workflow usage on reservoirnet for RC with two basic use-cases, and finally, in Section 4 we investigate the various challenges associated with an advanced case-study leveraging RC for forecasting COVID-19 hospitalizations. This case-study exploration includes detailed guidance on the modeling strategy, the selection of hyperparameters, and the implementation process.

# 2  RC presentation

RC is a machine learning paradigm which is most often implemented as Echo State Networks (ESNs), i.e. the firing-rate neuron version (Jaeger 2001). An ESN is described by three matrices of connectivity: an input layer $W_{in}$, a recurrent layer $W$ and an output layer $W_{out}$. At each time step, the input vector $u_t$ is projected into the reservoir which is also combined with reservoir past state $x(t-1)$ through

---

[2]In this article, we employ the term "train set" to refer to the combined dataset consisting of both the training and validation sets, which are cycled through in a cross-validation manner.

the recurrent connections. The output $y(t)$ is linearly read-out from the reservoir. Input $W_{in}$ and recurrent $W$ matrices are kept random; only the output matrix $W_{out}$ is trained in an offline or online method. Often a ridge regression (i.e. a regularized linear regression) is used to obtain the desired outputs $y(t)$ from the reservoir states $x(t)$. Figure 1 depicts the architecture. For simplicity, we will use the term "reservoir computing" for "Echo State Network" in the remainder of the paper.

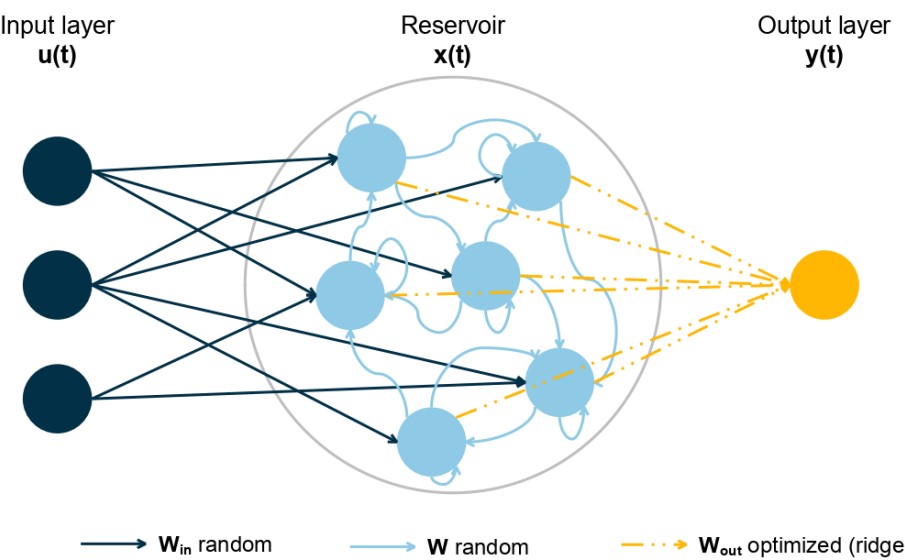

Figure 1: Reservoir computing is composed of an input layer, a reservoir and an output layer. Connection between input layer and reservoir and inside reservoir are random. Only the output layer is optimized based on a ridge regression. Adapted from Trouvain et al. (2020)

The input layer $u(t)$ is an $M$-dimension vector, where $M$ is the number of input time series, which corresponds to the values of the input time series at time $t$ where $t = 1, \dots, T$. The reservoir layer $x(t)$ is an $N_{res}$-dimensional vector where $N_{res}$ is the number of nodes in the reservoir. The value $x(t)$ is defined as follow:

$$x(t + 1) = (1 - \alpha)x(t) + \alpha \tanh\left(W x(t) + W_{in}u(t + 1)\right). \tag{1}$$

The leaking rate $\alpha \in [0, 1]$ defines the update rate of the nodes. The closer $\alpha$ is to 1, the more the reservoir is sensitive to new inputs $u(t)$. Therefore, the reservoir state at time $t + 1$ denoted $x(t + 1)$ depends on the reservoir state at the previous time $x(t)$ and the new inputs $u(t + 1)$. The function $tanh()$ represents the activation function, applied element-wise to each component of the vector, ensuring that each node's activation is scaled between $-1$ and 1. Both $W_{in}$ and $W$ are random matrices of size $N_{res} \times M$ and $N_{res} \times N_{res}$ respectively.

The input-reservoir connection matrix ($W_{in}$) and the intra-reservoir connection matrix ($W$) are generated in three steps. $W_{in}$ is generated using a Bernoulli (bimodal) distribution where each value can be either $-I_{scale}(m)$ or $I_{scale}(m)$ with an equal probability where $m = 1, \dots, M$ corresponds to a given feature in the input layer. The input scaling, denoted $I_{scale}$, is a hyperparameter coefficient common to all features from the input layer or specific to each feature $m$. In that case, the more important the feature is, the greater should be its input scaling. $W$ is generated from a Gaussian distribution $\mathcal{N}(0, 1)$. Both $W_{in}$ and $W$ then undergo sparsification, where a connectivity mask is applied to retain only 10% of the connections, enforcing sparsity. In a third step, the $W$ matrix is

scaled according to the defined spectral radius, a hyperparameter defining the highest eigen value of $W$.

The final layer is a linear regression with ridge penalization where the explanatory features are the reservoir state and the variable to be explained is the outcome to predict such that:

$$W_{out} = YX^T(XX^T + \lambda I)^{-1}.$$

Where x(t) and y(t) are accumulated in X and Y respectively such that:

$$X = \begin{bmatrix} x(1) \\ x(2) \\ ... \\ x(T) \end{bmatrix} \text{ and } Y = \begin{bmatrix} y(1) \\ y(2) \\ ... \\ y(T) \end{bmatrix}.$$

The parameter $\lambda$ is the ridge penalization which aims to prevent overfitting. Additionally, one can also connect the input layer to the output layer to the reservoir nodes. In that case, $X$ is the accumulation of both such that :

$$X = \begin{bmatrix} x(1), u(1) \\ x(2), u(2) \\ ... \\ x(T), u(T) \end{bmatrix} \text{ and } Y = \begin{bmatrix} y(1) \\ y(2) \\ ... \\ y(T) \end{bmatrix}.$$

Overall, there are four main hyperparameters to be chosen by the user: i) the leaking rate which defines the memory of the RC, ii) the input scaling which defines the relative importance of the features, iii) the spectral radius which defines the connections of the neurons inside the reservoir which in turn defines the degree of non-linear combination of features, and iv) the ridge penalization which controls the degree of overfitting. The choice of hyperparameters often requires the user to evaluate the performance of different combinations of hyperparameters on a validation set before selecting the optimal combination to forecast on the test set.

## 3 Usage workflow

In this section, we will cover the basics of reservoirnet use including installation, classification and regression. A more in depth description is provided in Section 4 with the covid-19 forecast use case.

### 3.1 Installation

reservoirnet is an R package making the Python module ReservoirPy easily callable from R using reticulate R package Ushey, Allaire, and Tang (2024). It is available on CRAN (see https://cran.r-project.org/package=reservoirnet) and can be installed using:

```
# Install reservoirnet package from CRAN
install.packages("reservoirnet")
```

Alternatively, it can also be installed from GitHub:

```
# Install reservoirnet package from GitHub
devtools::install_github(repo = "reservoirpy/reservoirR")
```

For reservoirnet to work, it will require Python version 3.8 or higher, along with the reservoirpy module which can be installed with the install_reservoirpy() function:

```
reservoirnet::install_reservoirpy()
```

Reservoir Computing (RC) is well suited to both regression and classification tasks. We will introduce a simple example for both task.

## 3.2 Package workflow overview

The workflow of reservoirnet is described in Figure 2. A reservoir model is created by the association of an input layer (a matrix), a reservoir, and an output layer. Both the reservoir and the output layer are created using the function `reservoirnet::createNode()` by specifying the node type (i.e., either `Reservoir` or `Ridge`).

This function accepts several arguments to specify the hyperparameters of the reservoir and will be detailed in future sections. After the reservoir and output layer are created, they can be connected using the %>>% operator, a specific pipe operator dedicated to reservoirnet. The model can then be fitted using `reservoirR_fit()` and used to make predictions on a new dataset using `predict_seq()`.

## 3.3 Basic regression use-case

### 3.3.1 Covid-19 data

In this first use-case, we will introduce the fundamental usage of the reservoirnet package. This demonstration will be conducted using the COVID-19 dataset that is included within the package. These data encompass hospitalization, positive RT-PCR (Reverse Transcription Polymerase Chain Reaction) results, and overall RT-PCR data sourced from Santé Publique France, which are publicly available on data.gouv.fr (for further details, refer to `help(dfCovid)`). Our primary objective is to predict the number of hospitalized patients 14 days into the future. To accomplish this, we will initially train our model on data preceding the date of January 1, 2022, and then apply it to forecast values using the following dataset.

We can proceed by loading useful packages - namely ggplot2 Wickham, Navarro, and Pedersen (2018) and dplyr Wickham et al. (2023), data and define the task:

```
# Load usefull packages
library(dplyr)
library(ggplot2)
library(reservoirnet)
# load dfCovid data from the reservoirnet package which contains Covid data
data("dfCovid")
# Set the forecast horizon to 14 days
dist_forecast = 14
# Set the train-test split to 2022-01-01
traintest_date = as.Date("2022-01-01")
```

Due to the substantial fluctuations observed in both RT-PCR metrics, our initial step involves applying a moving average computation over the most recent 7-day periods for these features. Additionally, we augment the dataset by introducing an `outcome` column and an `outcomeDate` column, which will serve as valuable inputs for model training. Moreover, we calculate the `outcome_deriv` as the difference between the outcome and the number of hospitalized patients (`hosp`), representing the variation in hospitalization in relation to the current count of hospitalized individuals. The resulting smoothed data is visualized in Figure 3.

```
dfOutcome <- dfCovid %>%
  # outcome at 14 days
  mutate(outcome = lead(x = hosp, n = dist_forecast),
         # Create a new column 'outcome' which contains the number of
         # hospitalizations ('hosp') shifted forward by 'dist_forecast' days
         # (14 days). This represents the outcome we want to predict.
```

Input layer :X

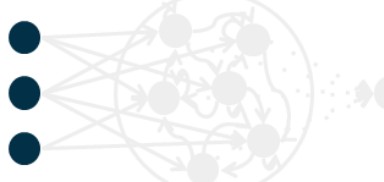

Instantiate reservoir :
```
reservoir <- createNode(nodeType = "Reservoir")
```

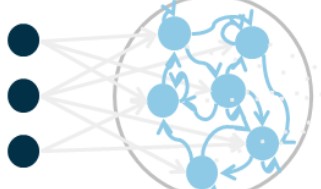

Instantiate output layer :
```
readout <- createNode(nodeType = "Ridge")
```

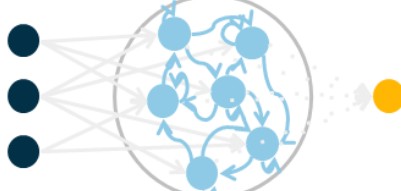

Build model :
```
model <- reservoir %>>% readout
```

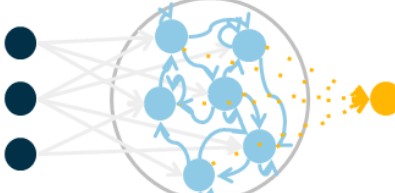

Fit model :
```
fit <- reservoirR_fit(node = model,
     X = X,
     Y = Y)
```

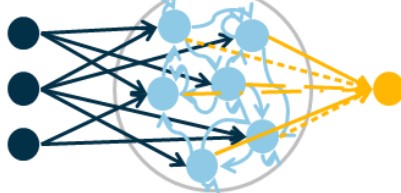

Forecast :
```
predict_seq(node = fit$fit, X = X)
```

Figure 2: Worflow of `reservoirnet`.

```
        outcomeDate = date + dist_forecast,
        # Create a new column 'outcomeDate' which is the current date plus the
        # forecast period (14 days).

        outcome_deriv = outcome - hosp) %>%
        # Create a new column 'outcome_deriv' which is the difference between
        # the predicted outcome and current hospitalizations.
        # This represents the change in hospitalizations over the forecast
        # period.

  # rolling average for tested and positive_pcr
  mutate_at(.vars = c("Positive", "Tested"),
            .funs = function(x) slider::slide_dbl(.x = x,
                                                  .before = 6,
                                                  .f = mean))
        # Apply a rolling mean (7-day average) to the 'Positive' and
        # 'Tested' columns.
        # The 'slider::slide_dbl' function is used to calculate the mean
        # over a window of 7 days (current day + 6 days before). This
        # smooths out daily fluctuations and provides a better trend
        # indicator.
```

Figure 3: Hospitalizations, number of positive RT-PCR and number of RT-PCR of Bordeaux University Hospital.

### 3.3.2 First reservoir

The objective of this task is to train a RC model using the input features to forecast the number of hospitalized patients 14 days ahead, as illustrated in Figure Figure 4.

Setting a reservoir is done with the createNode() function. The important hyperparameters are the following :

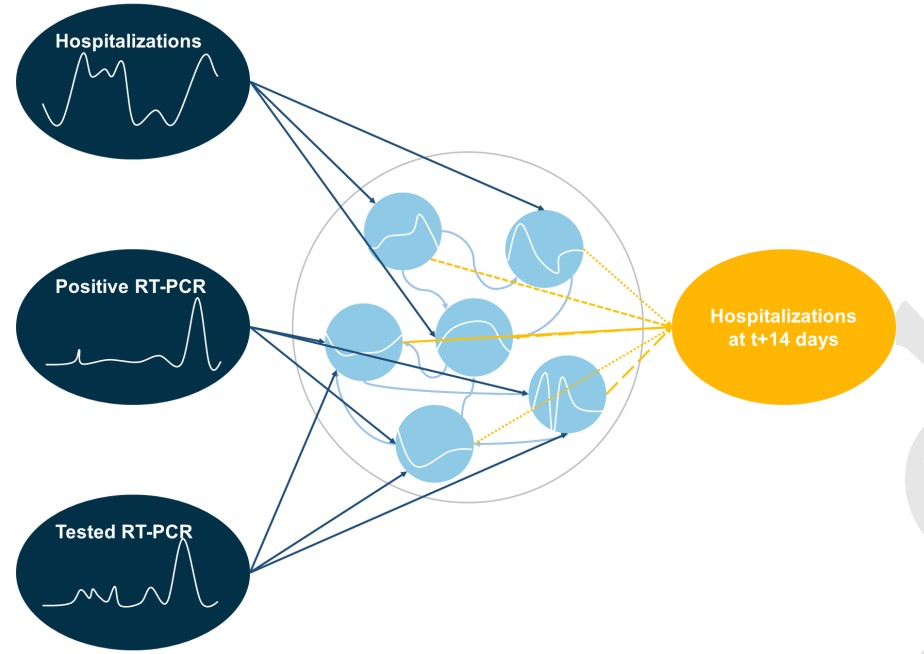

Figure 4: Regression use case: Forecasting the number of hospitalized patients 14 days ahead.

- Number of nodes (`units`) : it corresponds to the number of nodes inside the reservoir. Usually, the more the better, but more nodes increases the computation time.
- Leaking rate (`lr`) : the leaking rate corresponds to the balance between the new inputs and the previous state. A leaking rate of 1 only consider information from new inputs.
- Spectral radius (`sr`): the spectral radius is the largest eigenvalue in modulus of the reservoir connectivity matrix. A small spectral radius induces stable dynamics inside the reservoir, a high spectral radius induces a chaotic regime inside the reservoir.
- Input scaling (`input_scaling`): the input scaling is a gain applied to the input features of the reservoir.
- Warmup (`warmup`) : it corresponds to the number of time step during which the data are propagating into the reservoir but not used to fit the output layer. This hyperparameter is set in the `reservoirR_fit()` function.

In addition, we can set the seed (`seed`). Because the reservoir connections are set at random, setting the seed is a good approach to ensure reproducibility.

For this part of the tutorial, we will set the hyperparameter at a given value. Hyperparameter optimization will be detailed at Section 4.

```
# Create a reservoir computing node using the 'createNode' function from the
# reservoirnet package.
# Arguments:
# - nodeType = "Reservoir": Specify the type of node to be a reservoir.
# - seed = 1: Set the seed for reproducibility, ensuring consistent results
#             when the model is run multiple times.
# - units = 500: Set the number of reservoir units (neurons) to 500.
# - lr = 0.7: Set the leakage rate (lr) of the reservoir, which controls how
#             quickly the reservoir state decays over time.
# - sr = 1: Set the spectral radius (sr) of the reservoir, which influences the
#           stability and memory capacity of the reservoir.
```

```
# - input_scaling = 1: Set the input scaling factor, which scales the input
#                       signal before it is fed into the reservoir.

reservoir <- reservoirnet::createNode(nodeType = "Reservoir",
                                      seed = 1,
                                      units = 500,
                                      lr = 0.7,
                                      sr = 1,
                                      input_scaling = 1)
```

Then we can feed the data to the reservoir and see the activation state of the reservoir $x(t)$. To do so, we first prepare the data and transform it to a matrix.

```
## select explanatory features of the train set and transform it to an array
X <- dfOutcome %>%
  filter(outcomeDate < traintest_date) %>%
  select(hosp, Positive, Tested) %>%
  as.matrix()
```

Then we run the `predict_seq()` function. It takes as input a node (i.e a reservoir or a reservoir associated with an output layer) and the feature matrix.

```
# Generate the state of the reservoir using the 'predict_seq' function from the
# reservoirnet package.
# Arguments:
# - node = reservoir: The reservoir computing node created earlier.
# - X = X: The input data matrix containing the features 'hosp', 'Positive',
#          and 'Tested'.
# The function computes the state of the reservoir for each time step in the
# input sequence, effectively transforming the input data into the reservoir's
# high-dimensional state space.

reservoir_state <- predict_seq(node = reservoir, X = X)
```

Now we can visualize node activation using the `plot()` function presented at Figure 5 .

```
# Plot the reservoir state activation over time
plot(reservoir_state)
```

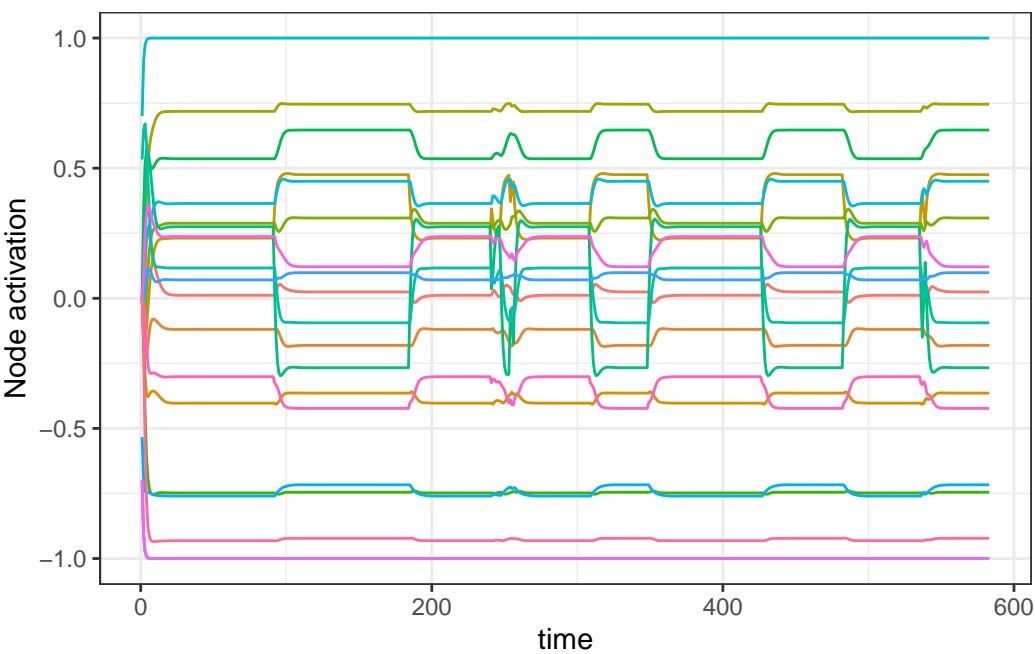

Figure 5: 20 random nodes activation over time.

Numerous nodes within the system exhibit a consistent equilibrium state. The challenge arises when the output layer attempts to extract knowledge from these nodes, as they do not convey meaningful information. This issue can be attributed to the disparate scales of the features. To address this concern, a practical approach involves normalizing the features by dividing each of them by their respective maximum values, thereby scaling them within the range of −1 to 1 by dividing by the maximum of the absolute value. Of note, here the features will be scaled between 0 and 1 because all features are positive.

```
# Standardise features by dividing by the maximum value can improve performance
# After standardisation, all features are on a similar scale which helps RC
stand_max <- function(x) return(x/max(abs(x)))
# scaled features
Xstand <- dfOutcome %>%
  filter(date < traintest_date) %>%
  select(hosp, Positive, Tested) %>%
  mutate_all(.funs = stand_max) %>%
  as.matrix() %>%
  as.array()
```

We then feed them to the reservoir and plot the node activation again. Compared to Figure 5, the obtained node activation at Figure 6 shows interesting trend outputs as no node seems saturated.

```
# feed the scaled features to the reservoir
reservoir_state_stand <- predict_seq(node = reservoir,
                                     X = Xstand,
                                     reset = TRUE)
# plot the output
plot(reservoir_state_stand)
```

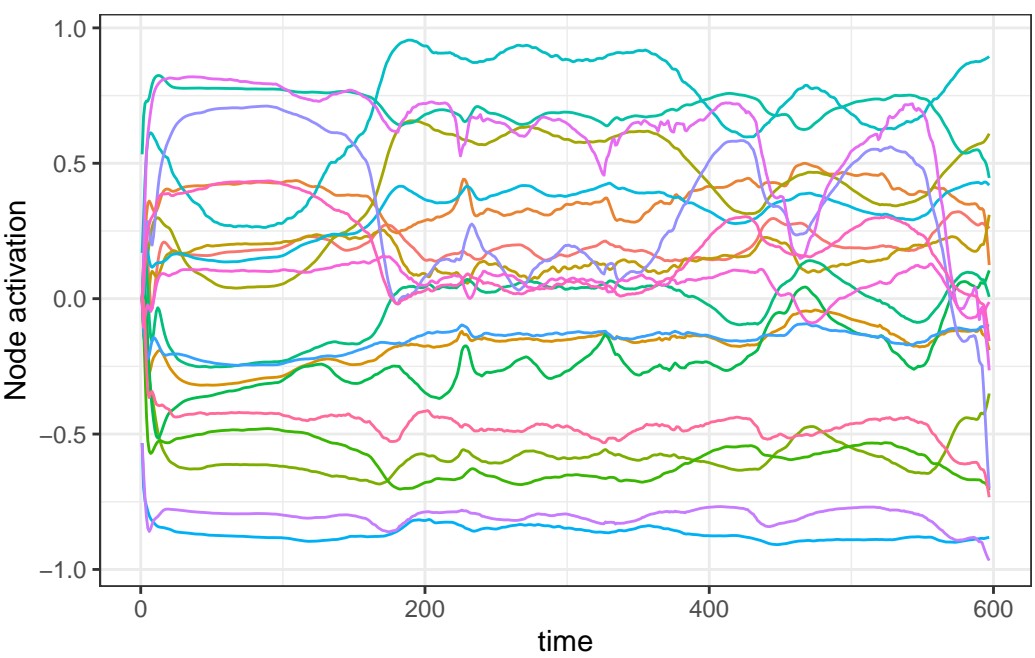

Figure 6: 20 random node activation over time. Scaled features.

### 3.3.3    Forecast

In order to train the reservoir, we should train the last layer which linearly combines the neuron's output.

#### 3.3.3.1    Set the ESN

Initially, we establish the output layer with the `createNode()` function, incorporating a ridge penalty set at `1e3`. It's important to note that this hyperparameter can be subject to optimization, a topic that will be explored in Section 4. This parameter plays a pivotal role in fine-tuning the model's conformity to the data. When set excessively high, the risk of underfitting arises, whereas setting it too low can lead to overfitting. We connect the output layer to the reservoir, with the `%>>%` operator, making the model ready to be trained.

```
readout <- reservoirnet::createNode(nodeType = "Ridge",
                                    ridge = 1e3)
# Create a readout node using ridge regression with the 'createNode' function
# from the reservoirnet package.
# Arguments:
# - nodeType = "Ridge": Specify the type of node to be a ridge regression
#                       readout.
# - ridge = 1e3: Set the regularization parameter (ridge) for the ridge
#                regression to 1000.
# Ridge regression is used to prevent overfitting by adding a penalty on the
# size of the coefficients.

model <- reservoir %>>% readout
# Link the reservoir and readout nodes to form a complete reservoir computing
# model. The '%>>%' operator connects the high-dimensional state generated by
```

```
# the reservoir to the readout layer, allowing the model to learn the mapping
# from the reservoir states to the target outputs.
```

 **3.3.3.2    Set the data**

 First we separate the train set on which we will learn the ridge coefficients and the test set on which
 we will make the forecast. We define the train set to be all the data before 2022-01-01 and the test
 data to be all the data to have forecast both on train and test sets.

```
# Perform some data management to isolate train and test sets
# train set
dftrain <- dfOutcome %>% filter(outcomeDate <= traintest_date)
yTrain <- dftrain %>% select(outcome)
yTrain_variation <- dftrain %>% select(outcome_deriv)
xTrain <- dftrain %>% select(hosp, Positive, Tested)
# test set
xTest <- dfOutcome %>% select(hosp, Positive, Tested)
```

 We standardize with the same formula as seen before. We learn the standardization on the training
 set and apply it on the test set. Then we convert the dataframe to matrix.

```
# copy train and test sets
xTrainstand <- xTrain
xTeststand <- xTest
# standardise based on training set values
ls_fct_stand <- apply(xTrain,
                      MARGIN = 2,
                      FUN = function(x) function(feature) feature/(max(x)))
lapply(X = names(ls_fct_stand),
       FUN = function(x){
         xTrainstand[,x] <<- ls_fct_stand[[x]](feature = xTrain[,x])
         xTeststand[,x] <<- ls_fct_stand[[x]](feature = xTest[,x])
         return()
       })
# convert to array
lsdf <- lapply(list(yTrain = yTrain,
                    yTrain_variation = yTrain_variation,
                    xTrain = xTrainstand,
                    xTest = xTeststand),
               function(x) as.matrix(x))
```

 **3.3.3.3    Train the model and predict**

 We then feed the reservoir with the train set using the `reservoirR_fit()` function. To do so, we set
 a `warmup` of 30 days during which the data are propagating into the reservoir but not used to fit the
 output layer.

```
### train the reservoir ridge output
fit <- reservoirnet::reservoirR_fit(node = model,
                                    X = lsdf$xTrain,
                                    Y = lsdf$yTrain,
                                    warmup = 30,
                                    reset = TRUE)
```

Now that the ridge layer is trained, we can forecast using the `predict_seq()` function. We set the parameter `reset` to `TRUE` in order to clean the reservoir from the data used by the training set.

```r
# Forecast with the trained reservoir on the test data
vec_pred <- reservoirnet::predict_seq(node = fit$fit,
                                      X = lsdf$xTest,
                                      reset = TRUE)

# Make figure to represent forecast on the train and test sets.

dfOutcome %>%
  mutate(pred = vec_pred) %>%
  na.omit() %>%
  ggplot(mapping = aes(x = outcomeDate)) +
  geom_line(mapping = aes(y = outcome,
                          color = "observed")) +
  geom_line(mapping = aes(y = pred,
                          color = "forecast")) +
  annotate("rect",
           xmin = traintest_date,
           xmax = max(dfOutcome$outcomeDate, na.rm = T),
           ymin = 0,
           ymax = max(dfOutcome$outcome, na.rm = T)*1.1,
           alpha = .2) +
  annotate("text", label = "Test set",
           x = as.Date("2022-08-01"), y = 2200, size = 7) +
  annotate("text", label = "Train set",
           x = as.Date("2021-03-01"), y = 2200, size = 7) +
  scale_color_manual(values = c("#3772ff", "#080708")) +
  theme_minimal() +
  labs(color = "", x = "Date", y = "Hospitalizations")
```

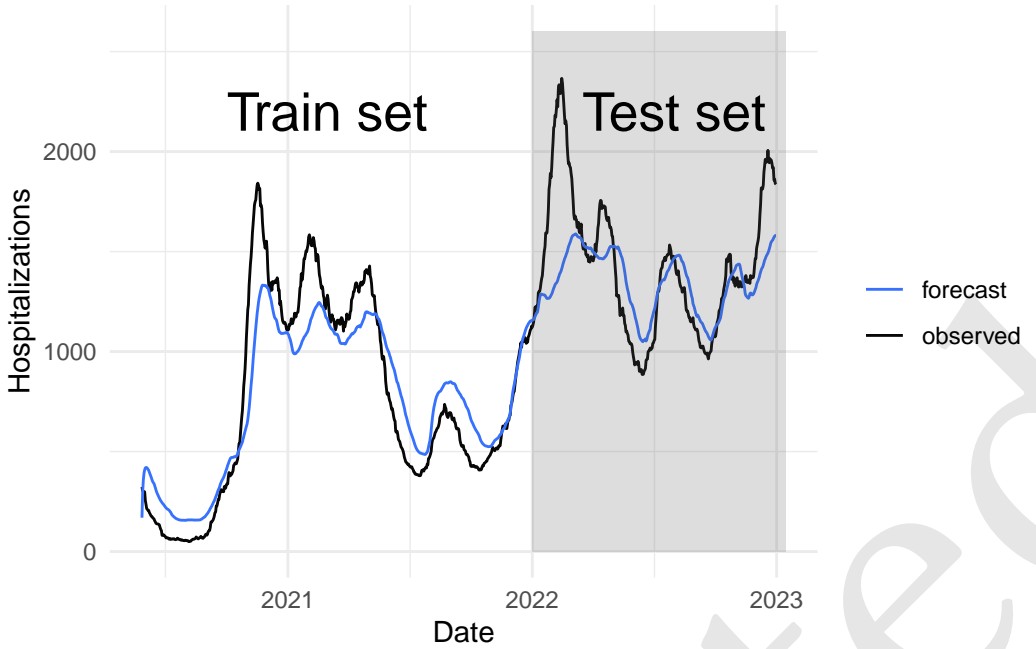

Figure 7: Forecast

221 We observe that the model forecast at Figure 7 is not fully accurate, both on the test set and the
222 train set. In that case, one option could be to reduce ridge penalization to fit more closely the data,
223 the optimization of ridge hyperparameter will be discussed at Section 4. Another possibility is to
224 ease the learning of the algorithm by forecasting the variation of the hospitalization instead of
225 the number of hospitalized patients. For that step, we will learn on the `outcome_deriv` contained
226 in `yTrain_variation` data which is defined outcome as `outcome_deriv = outcome - hosp`. As
227 depicted at Figure 8, this strategy improved the model forecast.

```r
## Fit reservoir on outcome variation instead of raw outcome
fit2 <- reservoirnet::reservoirR_fit(node = model,
                                     X = lsdf$xTrain,
                                     Y = lsdf$yTrain_variation,
                                     warmup = 30,
                                     reset = TRUE)
## Get the forecast on the test set
vec_pred2_variation <- reservoirnet::predict_seq(node = fit2$fit,
                                                 X = lsdf$xTest,
                                                 reset = TRUE)
## Transform the outcome variation forecast into hospitalization forecast
vec_pred2 <- vec_pred2_variation + xTest$hosp

## Plot the results
dfOutcome %>%
  mutate(Raw = vec_pred,
         Variation = vec_pred2) %>%
  tidyr::pivot_longer(cols = c(Raw, Variation),
                      names_to = "Outcome_type",
                      values_to = "Forecast") %>%
  na.omit() %>%
```

```r
ggplot(mapping = aes(x = outcomeDate)) +
geom_line(mapping = aes(y = outcome,
                        color = "observed")) +
geom_line(mapping = aes(y = Forecast,
                        color = "Forecast")) +
annotate("rect",
         xmin = traintest_date,
         xmax = max(dfOutcome$outcomeDate, na.rm = T),
         ymin = 0,
         ymax = max(dfOutcome$outcome, na.rm = T)*1.1,
         alpha = .2) +
annotate("text", label = "Test set",
         x = as.Date("2022-08-01"), y = 2200, size = 5) +
annotate("text", label = "Train set",
         x = as.Date("2021-03-01"), y = 2200, size = 5) +
facet_wrap(Outcome_type ~ .,
           labeller = label_bquote(cols = "Outcome" : .(Outcome_type))) +
scale_color_manual(values = c("#3772ff", "#080708")) +
theme_minimal() +
theme(legend.position = "bottom") +
labs(color = "", x = "Date", y = "Hospitalizations")
```

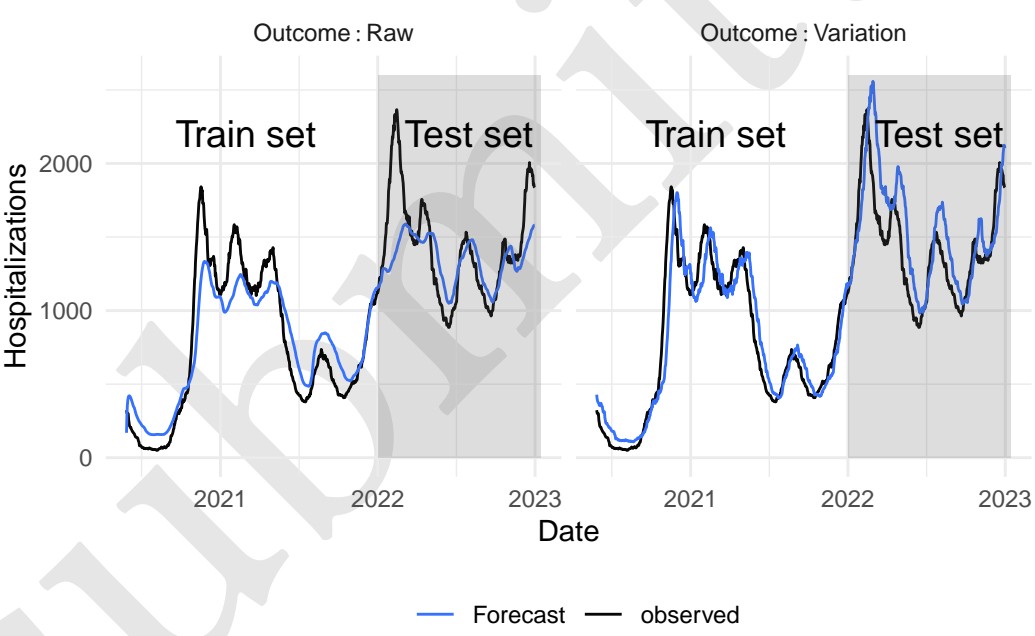

Figure 8: Covid-19 hospitalizations forecast. The model is either trained to forecast the number of hospitalizations (denoted Raw) or the variation of the hospitalizations compared to current level of hospitalisation (denoted Variation)

### 3.4 Classification

#### 3.4.1 The Japanese vowel dataset

This example is largely inspired from the classification tutorial of reservoirpy. To illustrate the classification task, we will use the Japanese vowel dataset (Kudo, Toyama, and Shimbo (1999)). The data can be loaded from `reservoirnet` as follow :

```
# Get the Japanese vowels dataset using the 'generate_data' function from the
# reservoirnet package.
# The dataset contains preprocessed features and labels for classification.
# Then we isolate train and test sets
japanese_vowels <- reservoirnet::generate_data(dataset = "japanese_vowels")[[1]]
X_train <- japanese_vowels$X_train
Y_train <- japanese_vowels$Y_train
X_test <- japanese_vowels$X_test
Y_test <- japanese_vowels$Y_test
```

The dataset comprises 640 vocalizations of the Japanese vowel æ, contributed by nine distinct speakers. Each vocalization represents a time series spanning between 7 and 29 time steps, encoded as a 12-dimensional vector denoting the Linear Prediction Coefficients (LPC). A visual representation of six distinct utterances from the test set, originating from three different speakers, is depicted in Figure 9.

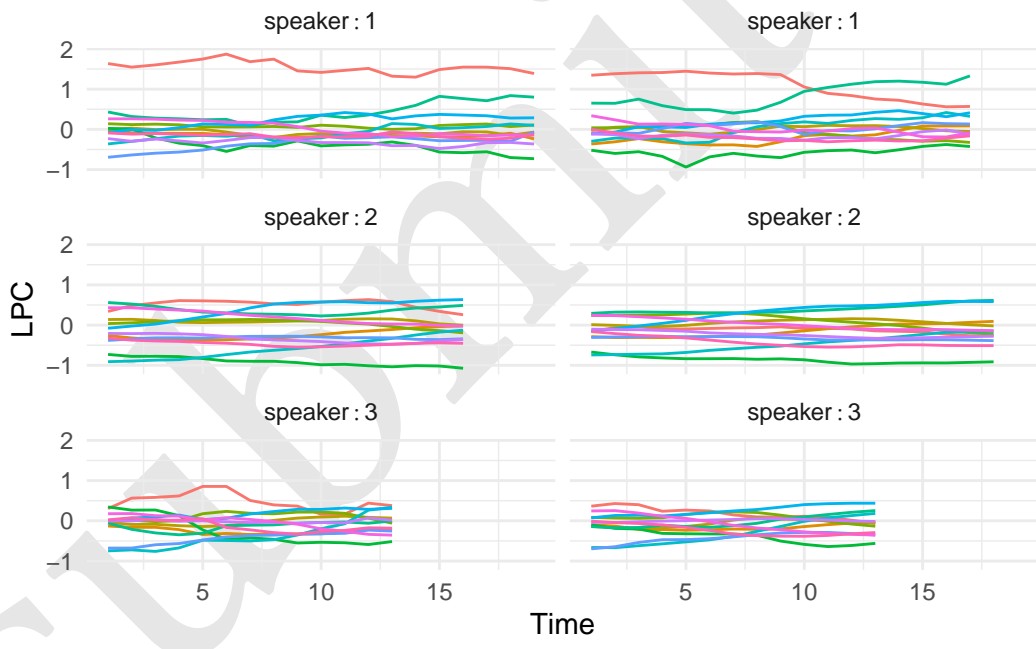

Figure 9: Vowel dataset, sample with 3 speakers and 2 utterance each.

The primary objective involves the attribution of each utterance to its respective speaker, this is denoted as classification or sequence-to-vector encoding. The secondary objective involves the attribution of each time step of each utterance to its speaker, this is denoted as transduction or sequence-to-sequence encoding. While this second approach may seem somewhat superfluous in this context, it could be useful, for example, in cases where multiple speakers take turns speaking, allowing us to identify which sequence belongs to each individual speaker. Figure Figure 4 illustrates this task.

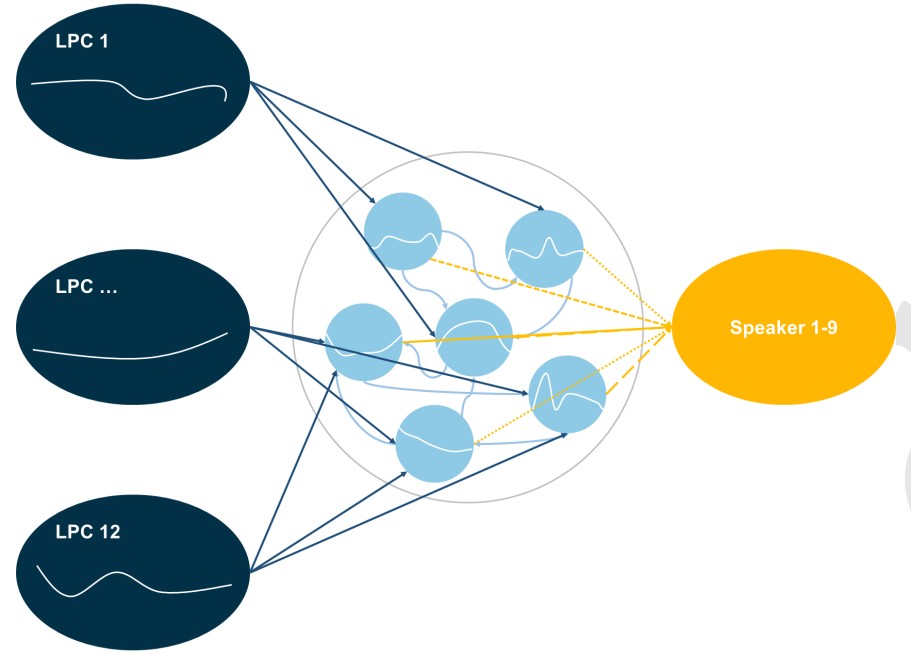

Figure 10: Classification use-case, identifying the speaker from an utterance.

### 3.4.2 Classification (sequence-to-vector model)

The first approach is the sequence-to-vector encoding. For this task we aim to predict the speaker of the whole utterance (i.e the label is assigned to the whole sequence). We first start by creating the reservoir and the output layer using `createNode()` function.

```
reservoir <- reservoirnet::createNode("Reservoir", units = 500,
                                      lr=0.1, sr=0.9,
                                      seed = 1)
# Create a reservoir computing node with 500 units using the 'createNode'
# function from the reservoirnet package.
# Arguments:
# - units = 500: Set the number of reservoir units (neurons) to 500.
# - lr = 0.1: Set the leakage rate (lr) of the reservoir to 0.1, controlling
#             how quickly the reservoir state decays over time.
# - sr = 0.9: Set the spectral radius (sr) of the reservoir to 0.9, influencing
#             the stability and memory capacity of the reservoir.
# - seed = 1: Set the seed for reproducibility, ensuring consistent results
#             when the model is run multiple times.
readout <- reservoirnet::createNode("Ridge",ridge=1e-6)
# Create a readout node using ridge regression with the 'createNode' function
# from the reservoirnet package.
# Arguments:
# - ridge = 1e-6: Set the regularization parameter (ridge) for the ridge
#                 regression to 1e-6.
# Ridge regression is used to prevent overfitting by adding a penalty on the
# size of the coefficients.
```

To perform this task, we need to modify the training and testing processes. Leveraging the inherent inertia of the reservoir, information from preceding time steps is preserved, effectively endowing the

RC with a form of memory. Consequently, the final state vector encapsulates insights gathered from all antecedent states. In the context of the sequence-to-vector encoding task, only the final state is used. To simplify this process, we introduce the `last_reservoir_state()` function, which extracts the final reservoir state. This process is executed as follows:

```r
states_train <- reservoirnet::last_reservoir_state(node = reservoir, X = X_train)
```

Then, we use only the final state for prediction. We first extract the final state using the `last_reservoir_state()` function and then use the trained readout to predict the vowel using the `predict_seq()` function with the `seq_to_vec` parameter set to `TRUE`:

```r
# Fit the reservoir using the last state vector (each observation is the whole
# vowel sequence)
res <- reservoirnet::reservoirR_fit(node = readout, X = states_train, Y = Y_train)
```

Then we can perform the prediction using only the final state. We first get the final state using the `last_reservoir_state()` function and use the trained readout to predict the vowel using the `predict_seq()` function with the `seq_to_vec` parameter set to `TRUE`.

```r
# The operation is repeated for the test set :
states_test <- reservoirnet::last_reservoir_state(node = reservoir, X = X_test)
Y_pred <- reservoirnet::predict_seq(node = readout, X = states_test, seq_to_vec = TRUE)
```

Figure 11 shows the prediction for the 6 utterances depicted at Figure 9 where the model correctly identifies the speaker.

```r
# A figure represents the performance on the test set
dfplotseqtovec <- lapply(vec_sample,
                  FUN = function(i){
                    speaker <- which(Y_test[[i]][1,] == 1)
                    Y_pred[[i]] %>%
                      as.data.frame() %>%
                      tidyr::pivot_longer(cols = everything(),
                                          names_to = "pred_speaker",
                                          values_to = "prediction") %>%
                      mutate(pred_speaker = gsub(x = pred_speaker,
                                                 pattern = "V", "")) %>%
                      mutate(speaker = speaker, .before = 1,
                             uterrance = i,
                             target = speaker == pred_speaker) %>%
                      return()
                  }) %>%
  bind_rows()

ggplot(dfplotseqtovec,
       mapping = aes(x = pred_speaker,
                     y = prediction,
                     fill = target)) +
  geom_bar(stat = "identity") +
  facet_wrap(uterrance ~ speaker,
             labeller = label_bquote(cols = "speaker" : .(speaker)),
             ncol = 2) +
  scale_fill_manual(values = c("#BDBDBD", "#A3CEF1")) +
```

```r
  theme_minimal() +
  theme(legend.position = "none") +
  labs(y = 'Score',
       x = "Speaker")
```

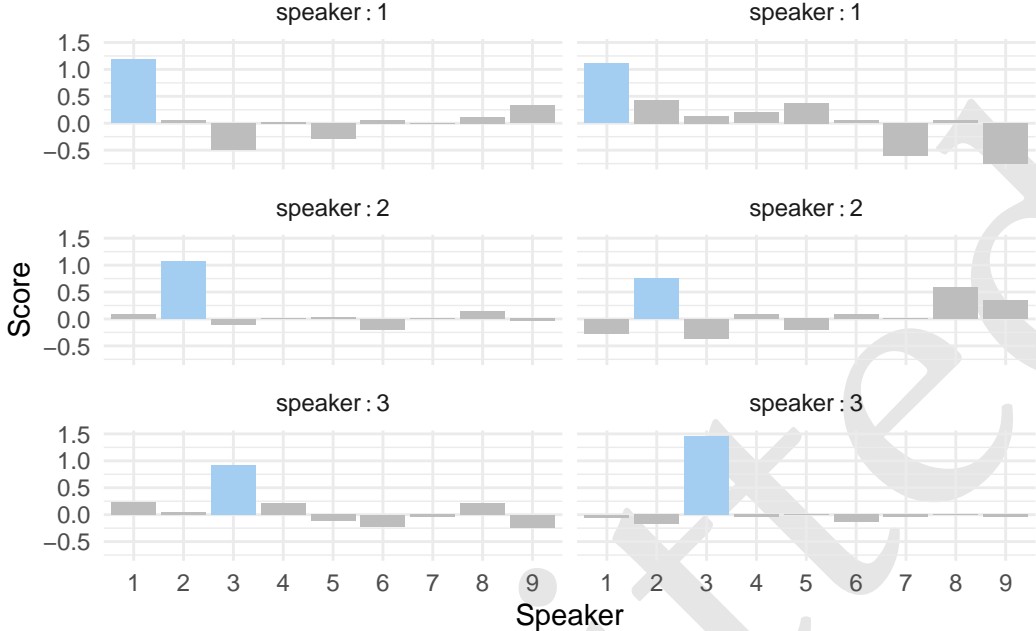

Figure 11: Prediction in a sequence-to-sequence approach 6 samples with 3 speakers and 2 utterance each. The speaker to predict is depicted in blue. For each of the 6 utterance, the model correctly identifies the speaker.

Then, we can also compute the overall accuracy :

```r
# The overall accuracy is evaluated
accuracy <- function(pred, truth) mean(pred == truth)

Y_pred_class <- sapply(Y_pred,
                       FUN = function(x) apply(as.matrix(x),1,which.max))
Y_test_class <- sapply(Y_test,
                       FUN = function(x) apply(as.matrix(x),1,which.max))

score <- accuracy(pred = Y_test_class, truth = Y_pred_class)

print(paste0("Accuracy: ", round(score * 100,3) ,"%"))
```

```
[1] "Accuracy: 92.703%"
```

### 3.4.3 Transduction (sequence-to-sequence model)

For this task, the goal is to predict the speaker for each time step of each utterance. The first step is to get the data where the label is repeated for each time step. This is easily done with the `repeat_targets` argument as follow :

```
# For this new task where we want to forecast for each time step (instead of each utterance)
# we start by getting the data in the appropriate format
# Then we split the train and test data
japanese_vowels <- reservoirnet::generate_data(
    dataset = "japanese_vowels",
    repeat_targets=TRUE)$japanese_vowels
X_train <- japanese_vowels$X_train
Y_train <- japanese_vowels$Y_train
X_test <- japanese_vowels$X_test
Y_test <- japanese_vowels$Y_test
```

Then we can train a simple Echo State Network to solve this task. For this example, we will connect both the input layer and the reservoir layer to the readout layer, which is performed by the %>>% operator. This direct connection between the input layer and the output layer can be particularly useful when the relationship between the input sequences and the output is partially linear, potentially improving performance. Section 4 will explore this aspect in more detail through the SARS-CoV-2 prediction task.

```
# Create an input, a reservoir and an output layers
source <- createNode("Input")
readout <- createNode("Ridge",ridge=1e-6)
reservoir <- createNode("Reservoir",units = 500,lr=0.1, sr=0.9, seed = 1)
# Connect the input layer to the reservoir and connect both the input layer and
# the reservoir to the output layer
model <- list(source %>>% reservoir, source) %>>% readout
```

We can then fit the model and predict the labels for the test data. The reset parameter is set to TRUE to remove information from the reservoir from the training process.

```
# Fit the RC model
model_fit <- reservoirnet::reservoirR_fit(node = model,
                                          X = X_train,
                                          Y = Y_train,
                                          warmup = 2)
# Predict with the fitted model
Y_pred <- reservoirnet::predict_seq(node = model_fit$fit,
                                    X = X_test,
                                    reset = TRUE)
```

From the Y_pred and Y_test we represent at Figure 12 the predictions for the same patients as in Figure 9.

```
# Make a graph with a label for each time of each utterance
dfplotseqtoseq <- lapply(vec_sample,
               FUN = function(i){
                   speaker <- which(Y_test[[i]][1,] == 1)
                   Y_pred[[i]] %>%
                     as.data.frame() %>%
                     tibble::rowid_to_column(var = "Time") %>%
                     tidyr::pivot_longer(cols = -Time,
                                         names_to = "pred_speaker",
                                         values_to = "prediction") %>%
                     mutate(pred_speaker = gsub(x = pred_speaker,
```

```r
                                         pattern = "V", ""),
                    speaker = speaker,
                    uterrance = i,
                    .before = 1) %>%
        return()
    }) %>%
  bind_rows()

ggplot(dfplotseqtoseq, mapping = aes(x = Time,
                                     y = pred_speaker,
                                     fill = prediction)) +
  geom_tile() +
  facet_wrap(uterrance ~ speaker,
             labeller = label_bquote(cols = "speaker" : .(speaker)),
             ncol = 2) +
  scale_fill_gradient2(low = "#8ECAE6", high = "#FB8500", mid = "#023047",
                       midpoint = 0) +
  theme_minimal() +
  labs(y = 'Predicted speaker',
       fill = "Prediction score")
```

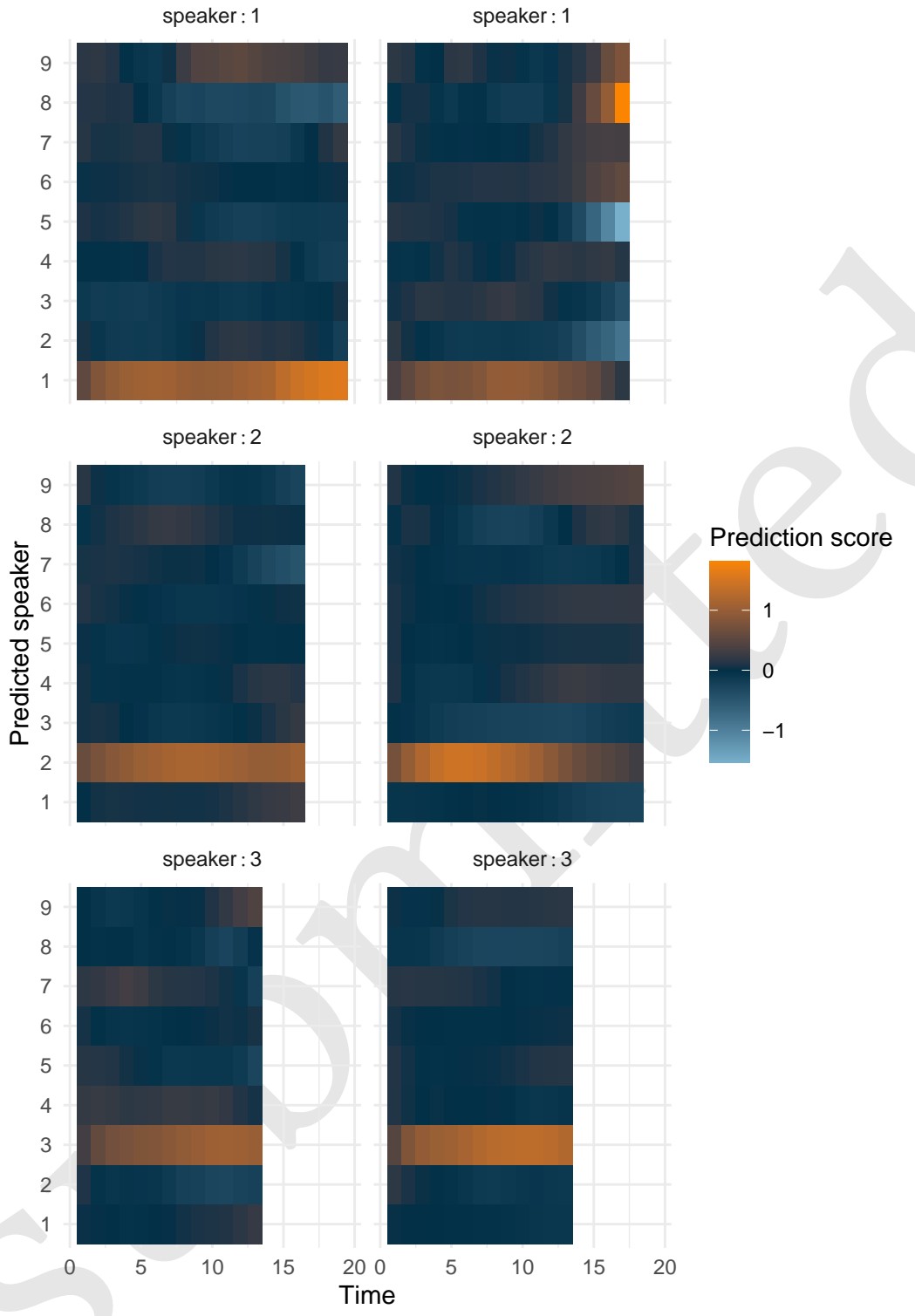

Figure 12: Prediction in a sequence-to-sequence approach 6 samples with 3 speakers and 2 utterance each. The higher the score of the speaker, the lighter the color.

For those 6 utterances, the model correctly identify the speaker for most of the time steps. We can then evaluate the overall accuracy of the model :

```
# Compute the accuracy
```

```
Y_pred_class <- sapply(Y_pred, FUN = function(x) apply(as.matrix(x),
                                                       1,
                                                       which.max))
Y_test_class <- sapply(Y_test, FUN = function(x) apply(as.matrix(x),
                                                       1,
                                                       which.max))
score <- accuracy(array(unlist(Y_pred_class)), array(unlist(Y_test_class)))

print(paste0("Accuracy: ", round(score * 100,3) ,"%"))
```

[1] "Accuracy: 92.456%"

# 4 Avanced case-study: Covid-19 hospitalizations forecast

## 4.1 Introduction

Since late 2020, millions of cases of SARS-CoV-2 infection have been documented across the globe (World Health Organisation 2020; COVID-19 Cumulative Infection Collaborators 2022; Carrat et al. 2021). This ongoing pandemic has exerted significant strain on healthcare systems, resulting in a surge in hospitalizations. This surge, in turn, necessitated modifications to the healthcare infrastructure and gave rise to population-wide lockdown measures aimed at preventing the saturation of healthcare facilities (Simões et al. 2021; Hübner et al. 2020; Kim et al. 2020). The capacity to predict the trajectory of the epidemic on a regional scale is of paramount importance for effective healthcare system management.

Numerous COVID-19 forecasting algorithms have been proposed using different methods (e.g ensemble, deep learning, mechanistic), yet none has proven entirely satisfactory (Cramer et al. 2022; Rahimi, Chen, and Gandomi 2021). In France, short-term forecasts with different methods have been evaluated with similar results (Paireau et al. 2022; Carvalho et al. 2021; Mohimont et al. 2021; Pottier 2021). In this context a machine learning algorithm based on linear regression with elastic-net penalization, leveraging both Electronic Health Records (EHRs) and public data, was implemented at Bordeaux University Hospital (Ferté et al. 2022). This model, which aimed at forecasting the number of hospitalized patients at 14 days, showed good performance but struggled to accurately anticipate dynamic shifts of the epidemic.

RC has been used in the context of covid-19 epidemic forecast (Kmet and Kmetova 2019; Liu et al. 2023; Ray, Chakraborty, and Ghosh 2021; Zhang et al. 2023; Ghosh et al. 2021). Among them, Ghosh et al. (2021), Liu et al. (2023) and Ray, Chakraborty, and Ghosh (2021) used it to forecast epidemic, Zhang et al. (2023) performed sentiment analysis and Kmet and Kmetova (2019) used it to solve optimal control related to vaccine. The evaluation of RC for epidemic forecast showed promising results in all approaches, being competitive with Long-Short Term Memory (LSTM) and Feed-Forward Neural Network (FFNN) in Ray, Chakraborty, and Ghosh (2021). However, the test period was short for Ghosh et al. (2021)} (21 and 14 days) and Ray, Chakraborty, and Ghosh (2021) (86 days) making it difficult to evaluate the behavior of the methods during epidemic dynamic shift. This was not the case for Liu et al. (2023) (6 months) but they implemented daily ahead forecast which would be difficult to use to manage a hospital. Finally, all three implementations used only one time series as input whereas it has been shown that using different data sources could improve forecast Ferté et al. (2022). Therefore, it is still difficult to assess the usefulness of RC over a large period and using many time series as inputs.

RC can be viewed as an extension of penalized linear regression, where inputs undergo processing by a reservoir, introducing the capacity for memory and non-linear combinations. Given the effectiveness

of penalized linear regression in COVID-19 forecasting, as highlighted in Ferté et al. (2022), and the promising results exhibited by RC in epidemic forecasting, as demonstrated in studies such as Ghosh et al. (2021), Liu et al. (2023), and Ray, Chakraborty, and Ghosh (2021), we have opted to employ RC for the prediction of hospitalizations at 14 days at the University Hospital of Bordeaux.

The aim of this study is to showcase the use of `reservoirnet` for an advanced use case in forecasting the SARS-CoV-2 pandemic in R. Several architectural choices will be evaluated, such as the connection between the input layer and the output layer, and the use of either individual input scaling per feature or a common input scaling. The performance of Reservoir Computing (RC) will be compared with elastic-net penalized regression (identified as the optimal model in Ferté et al. (2022)), while a more in-depth comparison of performance against other methods can be found in Ferté, Dutartre, Hejblum, Griffier, Jouhet, Thiébaut, Legrand, et al. (2024).

## 4.2 Methods

### 4.2.1 Data

The study utilized aggregated data spanning from May 16, 2020, to January 17, 2022, regarding the COVID-19 epidemic in France, drawing from various sources to enhance forecasting accuracy. These sources encompassed epidemiological statistics from Santé Publique France, weather data from the National Oceanic and Atmospheric Administration (NOAA), both providing department-level data (Smith, Lott, and Vose 2011; Etalab 2020) and Electronic Health Record (EHR) data from the Bordeaux Hospital providing hospital-level data. All data were daily updated. Santé Publique France data included information on hospitalizations, RT-PCR tests, positive RT-PCR results, variant prevalence, and vaccination data, categorized by age groups. NOAA data contributed temperature, wind speed, humidity, and dew point data, allowing for the computation of the COVID-19 Climate Transmissibility Predict Index (Roumagnac et al. 2021). EHRs data included hospitalizations, ICU admissions, ambulance service records, and emergency unit notes, with relevant COVID-19-related concepts extracted from the notes. Data are discussed more in depth in Ferté et al. (2022).

First derivative over the last 7 days were computed to enrich model information. To take into account measurement error and daily noise variation, data were smoothed using a local polynomial regression with a span of 21 days. As previously described, input features were scaled between -1 and 1 by dividing the observed value by the maximum of the absolute value of the given input feature.

All data are publicly available. Weather data can be obtained from Smith, Lott, and Vose (2011) using R package `worldmet` (Carslaw 2023). Vaccine data can be downloaded from Etalab (2020). EHRs data can be downloaded on dryad (Ferté et al. 2023). For privacy issues, publicly available EHRs data below 10 patients were obfuscated to 0. For convenience, all data were downloaded, merged and provided as replication material.

### 4.2.2 Evaluation framework

The task was to forecast 14 days ahead the number of hospitalized patients. As seen at Section 3.3, we will train the model to predict the variation of hospitalization, denoted as $hosp$, defined as $outcome_{t+14} = hosp_{t+14} - hosp_t$ with $t = 1, ..., T$. Metrics computation and visualizations will be performed on the predicted number of hospitalizations denoted as $\widehat{hosp_{t+14}} = \widehat{outcome_{t+14}} + hosp_t$.

The dataset was separated into two periods. First period from May 16, 2020 to March 1, 2021 served to identify relevant hyperparameters. Remaining data was used to evaluate the model performance.

The performance of the model was evaluated according to several metrics:

- the mean absolute error : MAE $= \frac{1}{T} \sum_{t=1}^{T} \left| \hat{hosp}_{t+14} - hosp_{t+14} \right|$.

- the median relative error : MRE = median $\left( \left| \frac{\hat{hosp}_{t+14} - hosp_{t+14}}{hosp_{t+14}} \right| \right)$.

- the mean absolute error to baseline : MAEB = $\frac{1}{T} \sum_{t=1}^{T} \left( \left| \hat{hosp}_{t+14} - hosp_{t+14} \right| - \left| hosp_t - hosp_{t+14} \right| \right)$.

- the median relative error to baseline : MREB = median $\left( \left| \frac{\hat{hosp}_{t+14} - hosp_{t+14}}{hosp_t - hosp_{t+14}} \right| \right)$

Median was chosen over mean for *MRE* and *MREB* because those metrics tend to have extremely high values when the denominator is close to 0 (i.e when the number of hospitalized patients is close to 0 or the number of patients hospitalized at 14 days is close to the current number of hospitalized patients respectively). *MAEB* and *MREB* compare model performance to a baseline model which predicts the current number of hospitalized patients at 14 days. Those metrics help to determine the information added by the model and is a good baseline as covid-19 forecast model do not always outperform this basic forecast (Cramer et al. (2022)).

Because the outcome is obfuscated below 10 hospitalizations for privacy reason, we set both the outcome and the forecast to 10 when the observed value was 0 or the forecasted value was below 10 when evaluating the model performance.

### 4.2.3 Models

We compared RC to elastic-net penalized regression (denoted as Enet). Furthermore we evaluated RC based on several architectures. First we compared RC with a single input scaling common to all features and a RC with a specific input scaling per feature. Second we compared RC where the input layer is connected to the output layer in addition to the connection between reservoir and output layer. Therefore, five models were evaluated :

- Elastic-net penalized regression denoted *Enet*
- RC with a single input scaling and no connection between input and ouput layers denoted *Common IS R %»% O*
- RC with a single input scaling and connection between input and ouput layers denoted *Common IS I+R %»% O*
- RC with multiple input scaling and no connection between input and ouput layers denoted *Multiple IS R %»% O*
- RC with multiple input scaling and connection between input and ouput layers denoted *Multiple IS I+R %»% O*

Because of the randomness of the reservoir, we took the median forecast of 10 reservoir on the train set to evaluate the performance of a given hyperparameter set. On the test set we aggregated the forecast of 40 reservoirs, each of them having one of the 40 best hyperparameter sets found on the train set. In addition, because covid-19 hospitalization is a non-stationary process, models were re-trained everyday using all previous days. To ease computation burden, only one day over two was used to find hyperparameters on the training set.

### 4.2.4 Hyperparameter optimisation using random search

RC relies mainly on 4 hyperparameters including the leaking rate (i.e "memory" parameter), spectral radius (i.e "chaoticity" parameter), input scaling (i.e "feature gain" parameter) and ridge (i.e penalization parameter). Input scaling can be either, common to all features or specific to each feature which increases the number of hyperparameter by the number of features.

Following the notation from `glmnet` package (Friedman, Hastie, and Tibshirani 2010), elastic-net penalized linear regression relies on two hyperparameters, lambda (i.e the penalization parameter) and alpha (i.e the compromise between lasso and ridge penalty)

Hyperparameter were selected in the training set (i.e before March 1, 2021) using a wrapper approach and a random search sampler using 2000 samples for each model. The sampling distribution were defined as follow :

- (RC) ridge and (Enet) lambda : log-uniform law defined between 1e-10 and 1e5
- (RC) input scaling and spectral radius : log-uniform law defined between 1e-5 and 1e5
- (RC) leaking rate : log-uniform law defined between 1e-3 and 1
- (Enet) alpha : uniform defined between 0 and 1

We provided large search space for all hyper-parameters. Search space was slightly reduced for leaking rate based on previous results and because a leaking rate of 1e-3 already imply that new inputs make the reservoir change really slowly which is not inline with the dynamic of covid-19 but would be appropriate for an application where the phenomena to forecast has a slow dynamic.

Finally, we provided an additional Enet model similar to the one in Ferté et al. (2022) where alpha was set to 0.5 and lambda was re-evaluated everyday in the test set based on previous data using the cross-validation procedure provided by `glmnet`.

## 4.3 Results

The goal of this task is to predict 14 days ahead the hospitalization. Figure 13 shows both the training set (i.e before 2021-03-01) and the test set where the blue curve correspond to the input features (first derivatives are not shown) and the orange curves correspond to the outcome the model is trained on (i.e the hospitalization variation) and the hospitalizations at 14 days on which the performance metrics are computed. The figures outline that the relation between the input features and the outcome evolve over time and that the time series is not stationary. For instance IPTCC (*Index PREDICT de Transmissivité Climatique de la COVID-19*) seems correlated to the outcome except that it completely miss the summer 2021 increase.

### 4.3.1 Hyperparameter selection

Figure 14 shows the hyperparameter optimisation using random search for the different RC architectures. We observe that model with multiple input scaling achieved better performance on the train set compared to model with single input scaling which is expected as they can adapt more closely to the data thanks to specific input scaling for each feature.

As expected, we observe that the optimal leaking rate is above 1e-2 for all RC which is coherent with the short term dynamic of covid-19 epidemic. Trends for other hyperparameters are less clear even though best hyperparameters sets were close for RC with common input scaling and for RC with multiple input scaling.

Figure 15 shows the hyperparameter search for RC with multiple input scaling and connected input layer. We observe that the random search tends to favor high importance given to derivative of positive RT-PCR (including the elderly) and the derivative of IPTCC. The remaining features do not exhibit a clear pattern.

### 4.3.2 Forecast performance

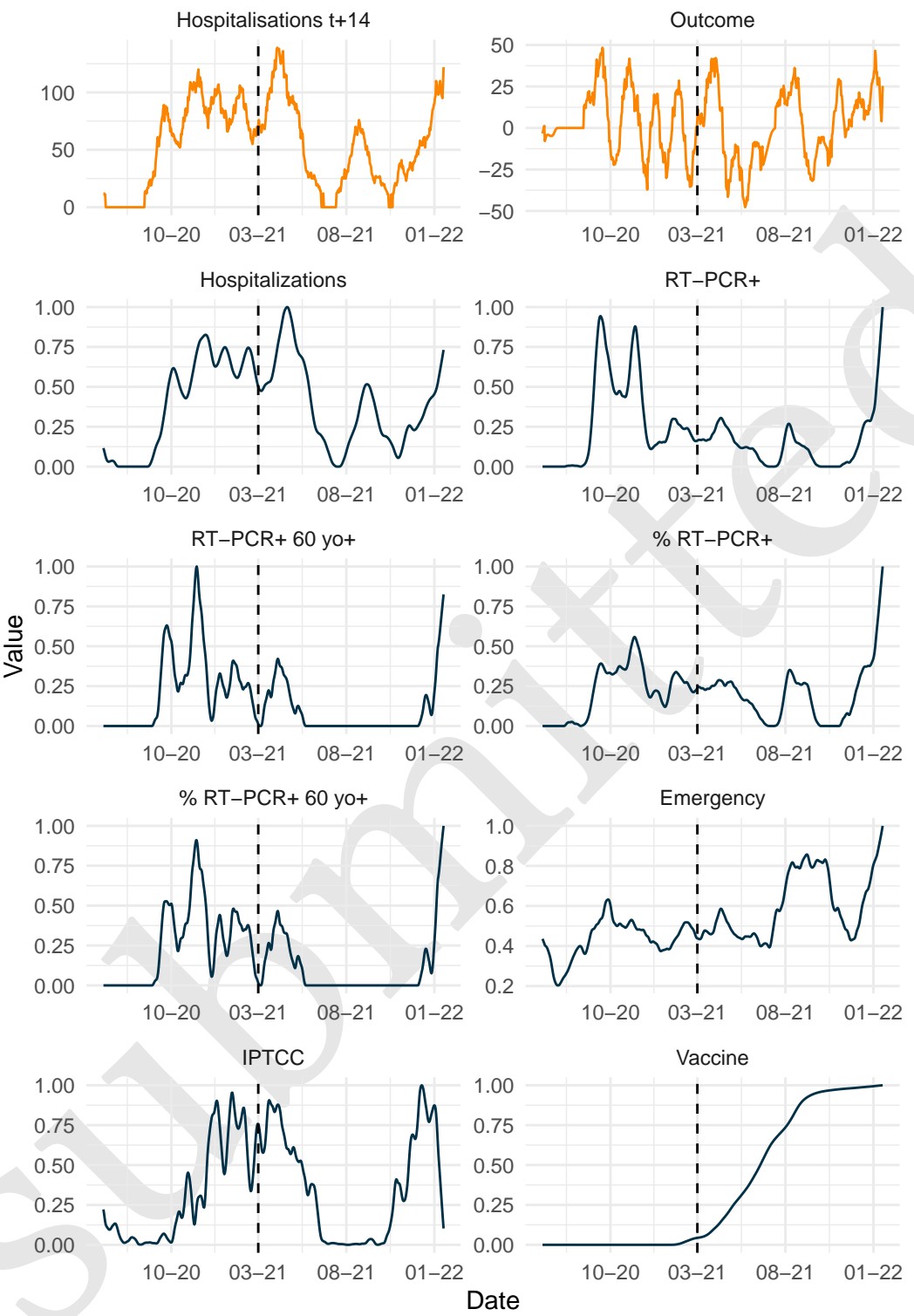

Figure 13: Covid-19 epidemic at BUH. Outcome of interest is presented in orange. Model is trained to forecast Outcome curve which corresonds to the difference between Hospitalisatiosn at 14 days and current hospitalisations. Other features are scaled (divide by the maximum of the feature) represented in darkblue.

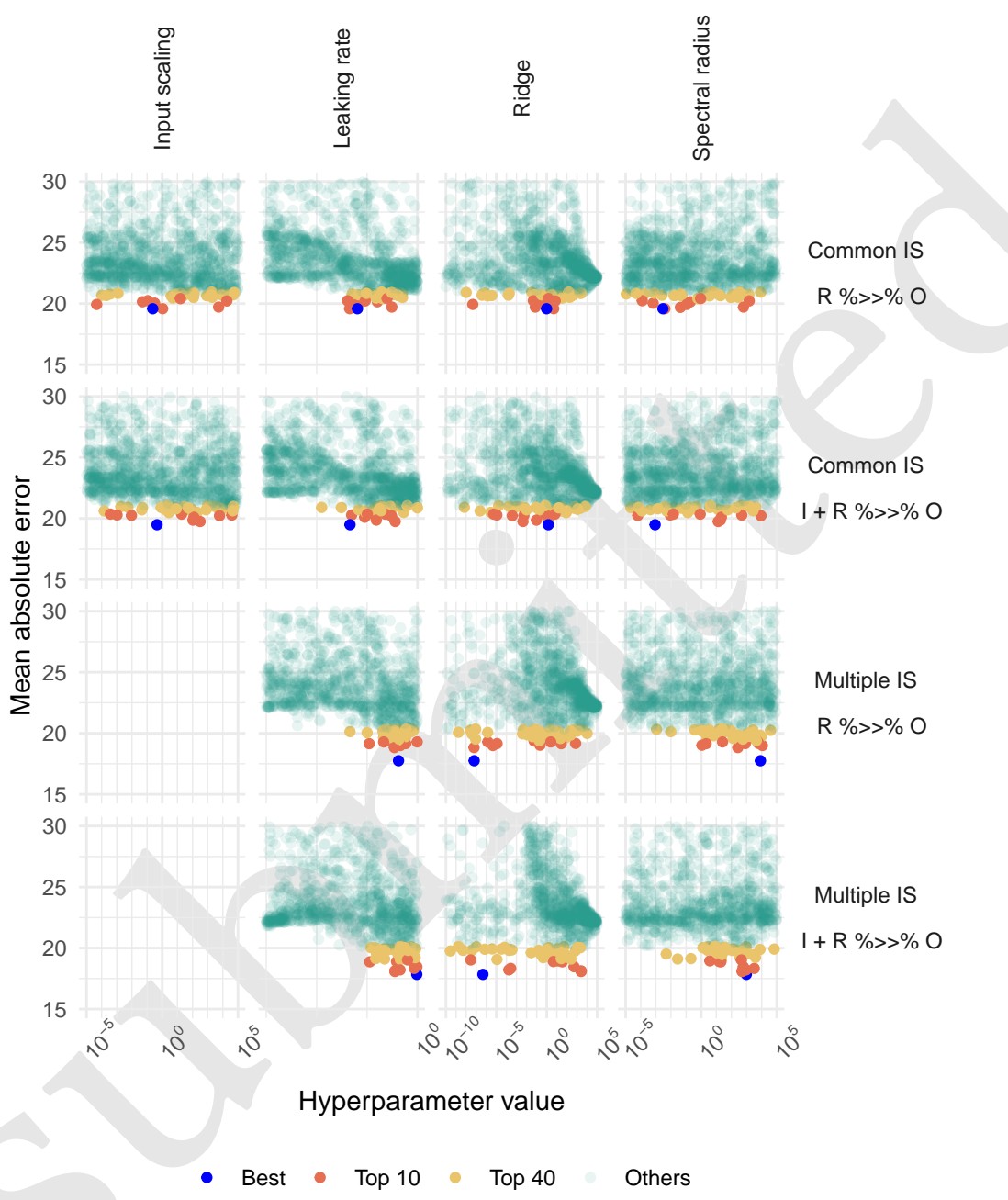

Figure 14: Hyperparameter evaluation on training set by random search. Hp sets with MAE above 30 were removed for clarity of visualisation.

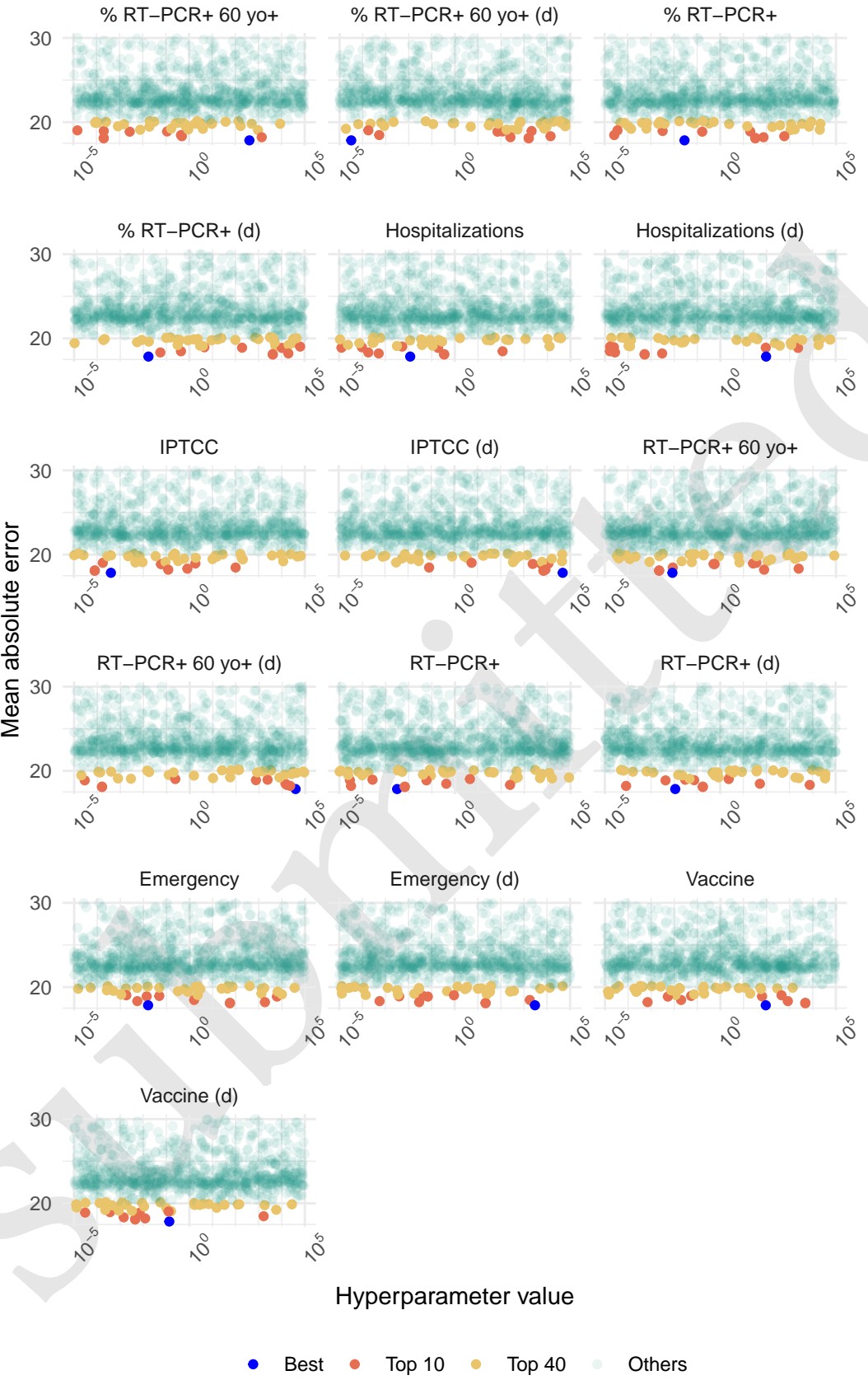

Figure 15: Hyperparameter evaluation on training set by random search of the model with multiple input scaling and no connection between input layer and output layer. Hp sets with MAE above 30 were removed for clarity of visualisation.

Table 1: Model performance with several reservoir configuration. For each setting, 40 reservoirs are computed and the forecast is the median of the 40 forecasts. Results show the performance metrics : MAE = Mean Absolute Error, MRE = Median Relative Error, MAEB = Mean Absolute Error to Baseline, MREB = Median Relative Error to Baseline.

Table 1: Model Performance

| Model | MAE | MRE | MAEB | MREB |
|---|---|---|---|---|
| Common IS: R %»% O | 15.23 | 0.26 | -3.50 | 0.85 |
| Common IS: I + R %»% O | 14.84 | 0.26 | -3.89 | 0.83 |
| Multiple IS: R %»% O | 15.38 | 0.28 | -3.35 | 0.82 |
| Multiple IS: I + R %»% O | 15.25 | 0.28 | -3.49 | 0.83 |
| Elastic-net | 16.40 | 0.29 | -2.34 | 0.93 |

Table 1 shows the performance on the test set. Best model according to all metrics was RC with common input scaling and connection between input and output layers. Having one input scaling per feature did not improve the model which might be due to low generalisability of the hyperparameter of the training set to the test set due to non-stationarity. Additionaly, connecting input layer to output layer improved the model forecast. All RC models performed better than the elastic-net model.

Figure 16 shows the forecast of the different models. We note that models struggle to accurately forecast slope shifts. For instance, summer 2021 initial increase is partially predicted by all models but its decrease is not well predicted. Winter 2021 increase is anticipated by all models but they tend to overestimate it because of the rise of vaccine effect.

### 4.3.3   Number of model to aggregate

Figure 17 show the individual forecast for the 40 best sets of hyperparameters of each RC architecture. Due to the internal random connection of the reservoir, we observe forecast stochasticity and relying on only one forecast is unreliable. We explored the number of model needed at Figure 18 which shows that after 10 models, forecast is stable and even 5 models for the simpler model with common input scaling which rely on less hyperparamters.

### 4.3.4   Input feature importance

We compared the coefficients of the output layer estimated for the input layer and the reservoir nodes. Additionally, we compared the coefficient given to the input layer by the output layer in the reservoir and the coefficient estimated by the elastic-net model.

Figure 19 illustrates the ranking of input layer compared to all connections to the output layer, including the 500 reservoir nodes and the 16 features of the input layer (excluding bias). The figure shows that the model with common input scaling tends to assign less weight to input layer compared to the model with multiple input scaling. This suggests that the reservoir with common input scaling provides more information than the reservoir with multiple input scaling, which aligns with its better performance, as shown in Table Table 1.

Furthermore, Figure 20 compares the coefficients assigned to input features by the elastic-net model and the RC models. While the coefficients are generally consistent across RC models, there are some notable differences with elastic-net. Specifically, certain features deemed important by the elastic-net model, such as the derivative of RT-PCR, and the derivative of Vaccine, are less important

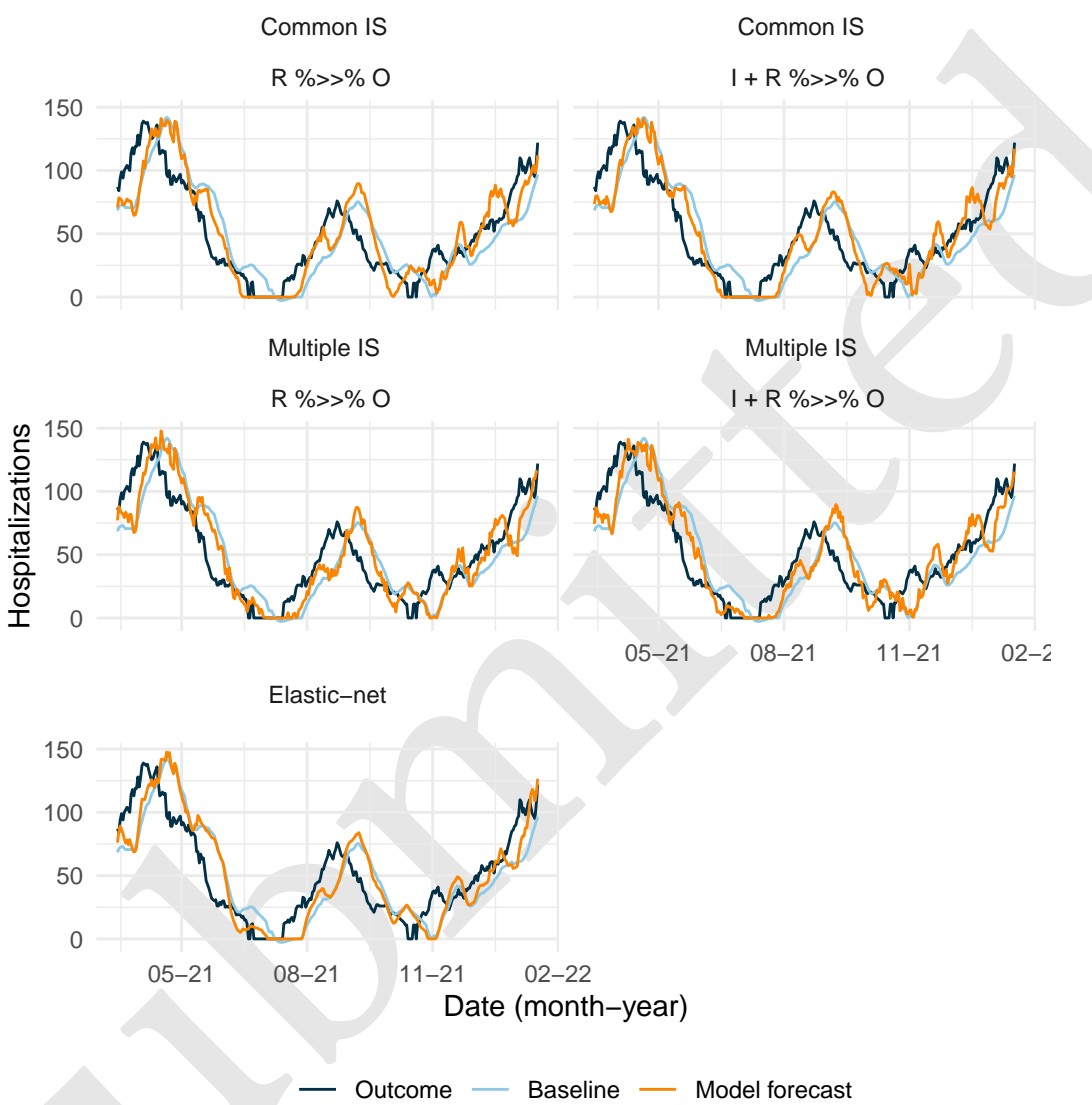

Figure 16: Reservoir computing forecast depending on the setting with and without monthly update. Red line is the median forecast of 40 reservoirs. Grey lines are individual forecast of each of the 40 reservoirs.

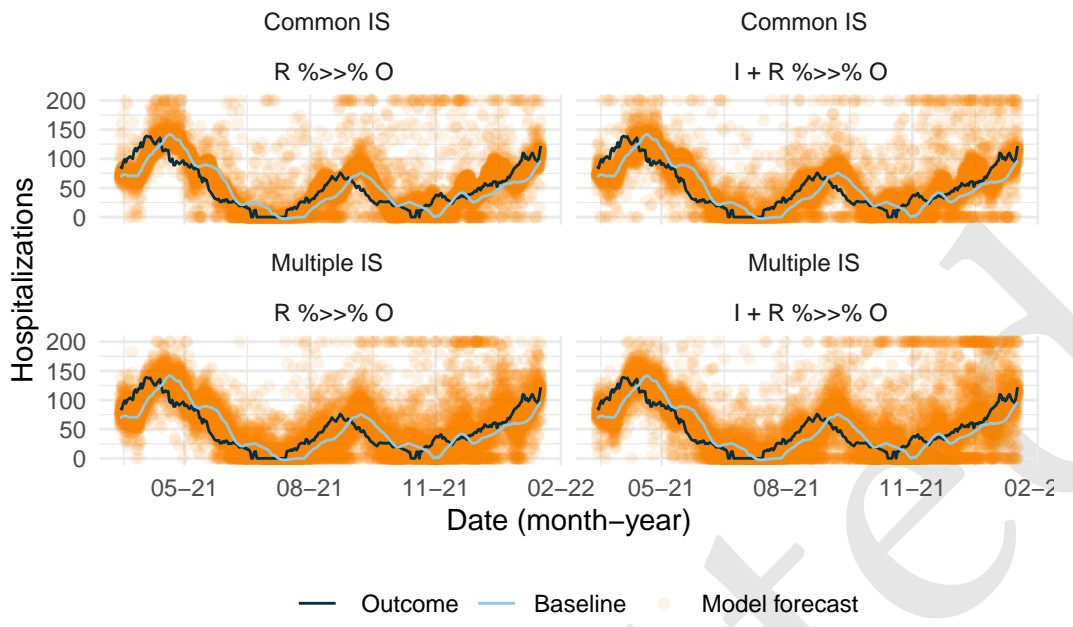

Figure 17: Individual forecast the 40 best hyperparameter sets for the different RC configuration. Forecast value above 200 were set to 200 for clarity.

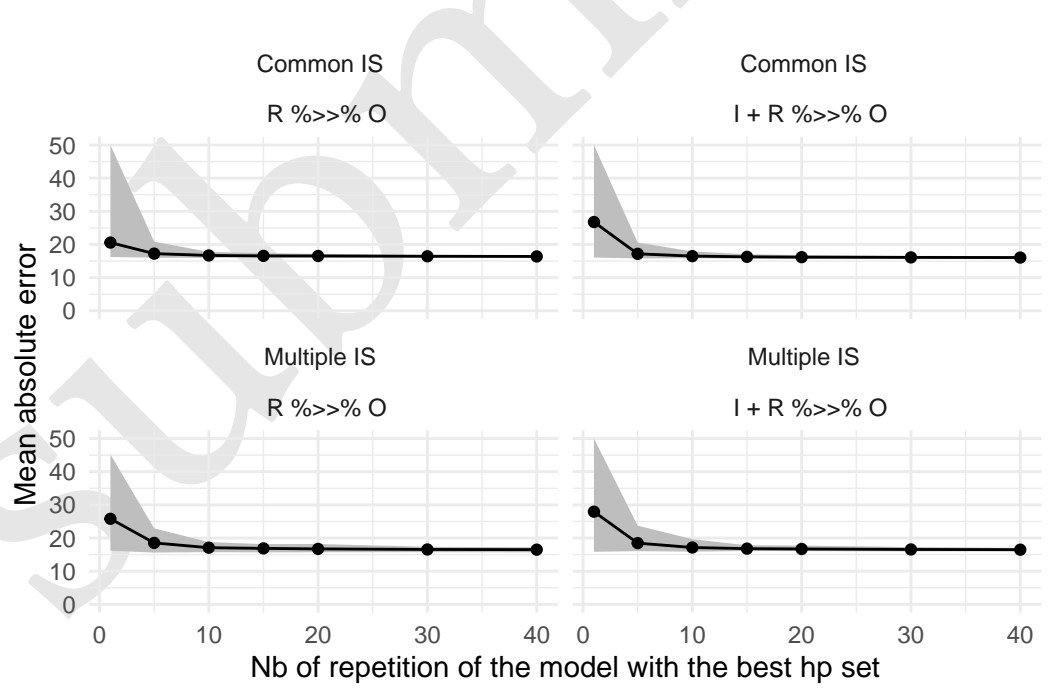

Figure 18: Mean absolute error depending on the number of aggregated reservoir.

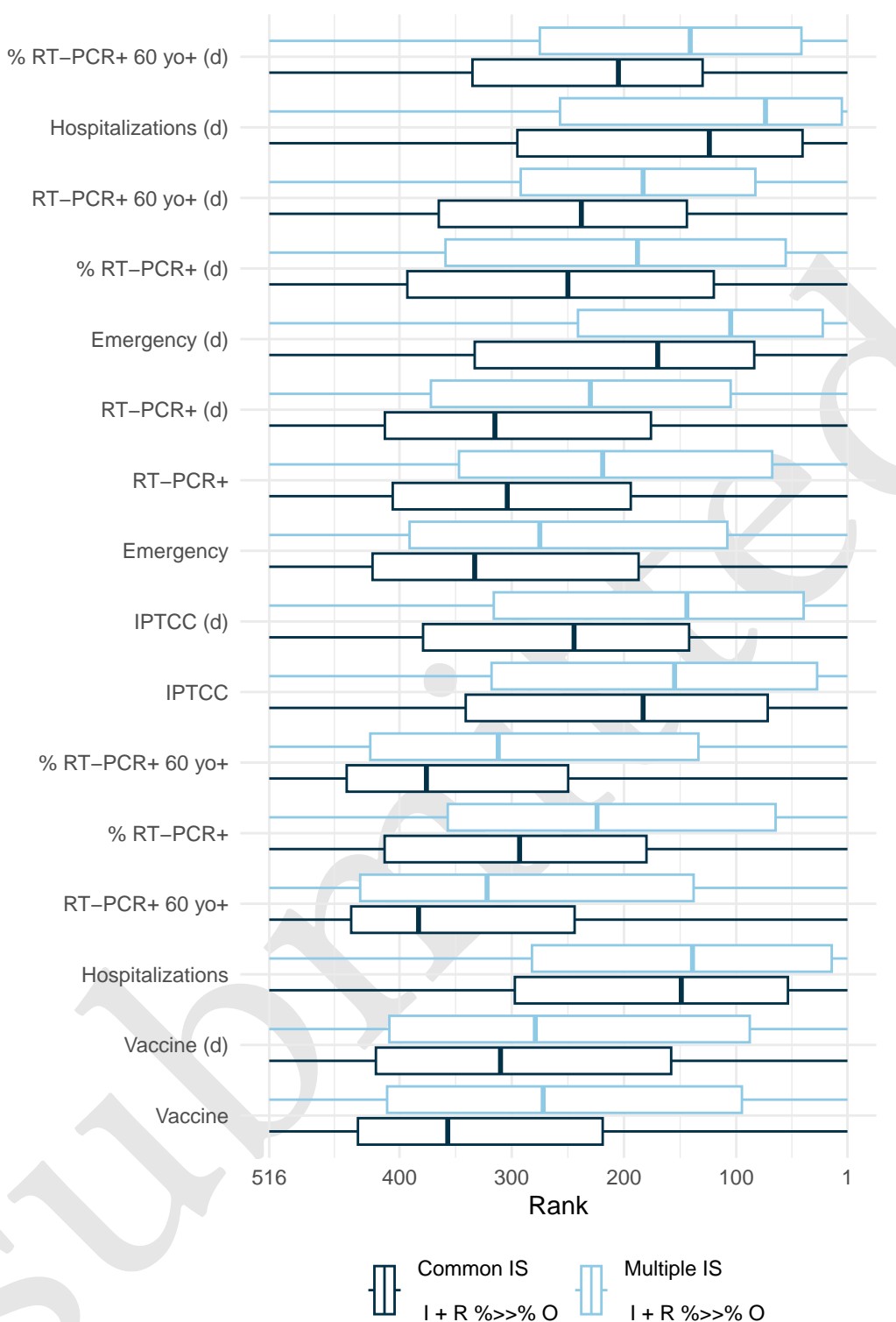

Figure 19: Mean feature importance of the 40 best hyperparameter sets by model, focus on the connection between the input and output layers. Models with direct connection between input and output layer are included. The rank is obtained by comparing the feature input layer and all other connection coefficients (both input and reservoir corresponding coefficients) attributed by the output layer at each date for each hyperparameter set. The higher the output layer's coefficient for the input layer, the closer its rank will be to 1 and the more important the feature is.

for the reservoir computing model. This may indicate that these features predictive ability is better conveyed by their relationship with other features, which is captured by the reservoir computing model but might not be by the elastic-net model. Conversely, emergency, IPTCC, proportion of positive RT-PCR, and hospitalizations are more important for the reservoir computing model than for the elastic-net model.

## 4.4  Discussion

In this specific application, we have demonstrated that RC exhibits commendable performance in comparison to Elastic-net, which serves as the reference model. Furthermore, we highlight the inherent challenges in forecasting within this context, primarily stemming from the non-stationarity of the time series.

All computations in this study were conducted using the `reservoirnet` package, and the entire codebase is accessible on Zenodo (Ferté, Ba, et al. 2024). This `R` package demonstrates its efficacy in implementing various reservoir architectures, including connection between the input layer and the output layer, as well as the utilization of several input scaling, all within the context of a real-world use case.

Given the substantial number of hyperparameters involved, we acknowledge that random search may not be the most efficient optimization algorithm. We have retained this approach for the sake of simplicity in this tutorial paper; however, meta-heuristic approaches, particularly those utilizing evolutionary algorithms, may prove more efficient, especially when employing multiple input scaling (Bala et al. 2018; Ferté, Dutartre, Hejblum, Griffier, Jouhet, Thiébaut, Hinaut, et al. 2024).

This study represents a novel contribution to epidemic forecasting utilizing RC. Notably, previous literature predominantly focused on simpler problems characterized by fewer input features or shorter evaluation periods (Liu et al. 2023; Ray, Chakraborty, and Ghosh 2021; Ghosh et al. 2021). Our findings underscore the potential of this approach for future epidemics, suggesting its potential to surpass more traditional epidemiological tools while maintaining a lightweight model structure compared to other RNNs.

It is worth noting that all models, including those presented in Ferté et al. (2022), face challenges in accurately predicting slope shifts, highlighting the need for further investigation. Specifically, additional work is required to extend the application of Reservoir Computing (RC) to high-dimensional settings, building upon insights gained from models that use a more extensive feature set. While RC has demonstrated promising performance for epidemic forecasting in high-dimensional settings, this task remains challenging (Ferté, Dutartre, Hejblum, Griffier, Jouhet, Thiébaut, Legrand, et al. 2024).

# 5  Discussion and conclusion

In this paper, we introduce the `R` package `reservoirnet`, which serves as a versatile tool for implementing reservoir computing based on `ReservoirPy`'s `Python` library. It offers flexibility in defining the reservoir architecture, including options for specifying connections between the input layer and the output layer, as well as variations in input scaling as demonstrated on a real-world use case.

We provided a comprehensive overview of the basic usage of the `reservoirnet` package through illustrative examples in regression and classification tasks. This introductory section serves as a foundation for `R` users, offering step-by-step guidance on constructing and training reservoir computing models using the package. By demonstrating the application of RC in both regression and classification scenarios, we aim to equip users with the essential knowledge and skills needed to harness the capabilities of reservoir computing for diverse tasks.

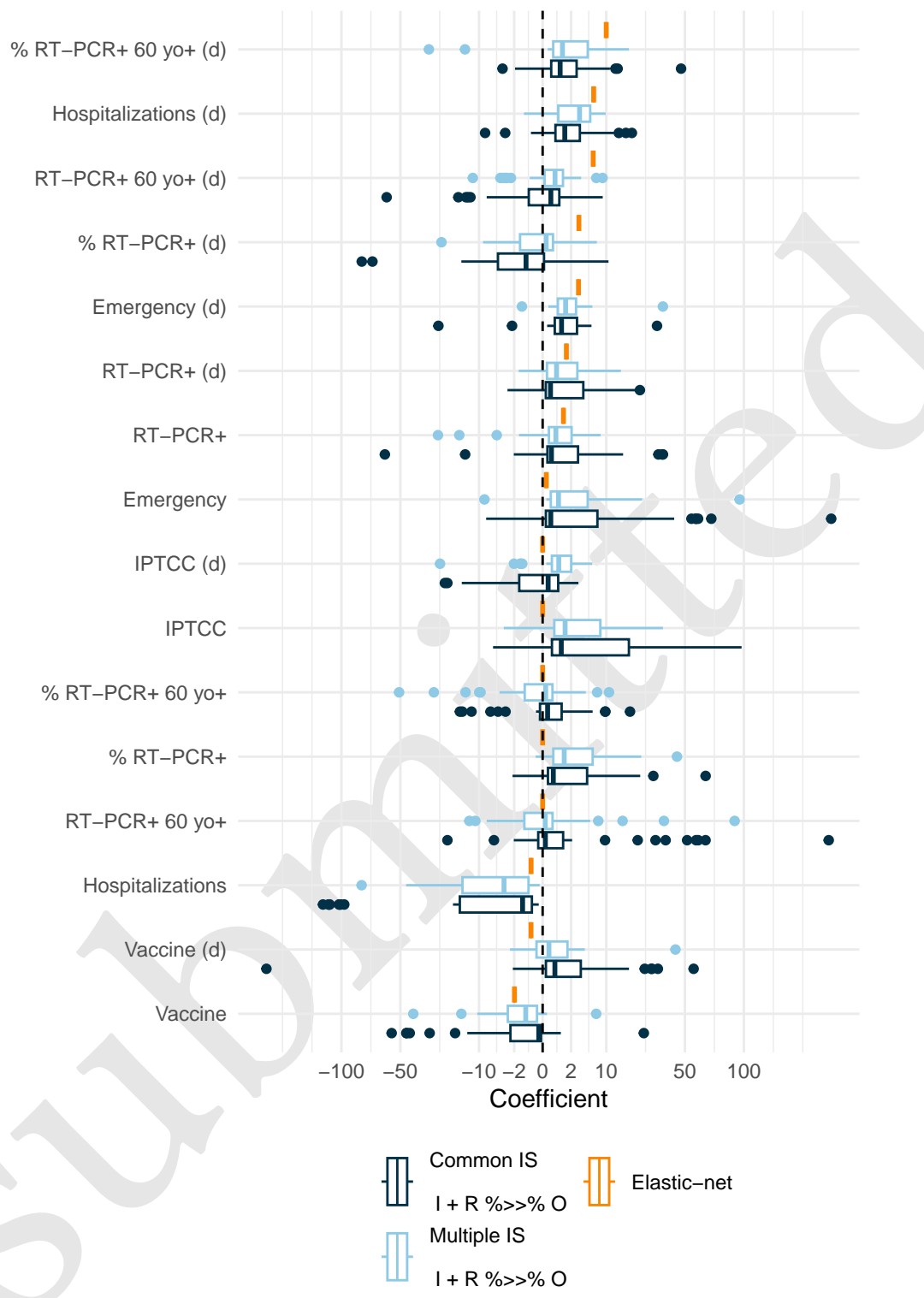

Figure 20: Mean feature coefficient of the 40 best hyperparameter sets by model and the elastic-net model. Only models with direct connection between input and output layer are included. The coefficients were calculated as the average value across all dates for each feature, model and hyperparamet set.

Drawing on the robust foundation of the `ReservoirPy` structure, a well-maintained `Python` library, this package inherits its reliability and longevity. We have focused on providing access to the fundamental features, building upon the strong base provided by `ReservoirPy`. Therefore, this initial version of `reservoirnet` must evolve in tandem with the growing understanding and adoption of RC within the `R` community.

# Acknowledgements

We thank Romain Griffier for his expertise on the Bordeaux University Hospital Data.

Experiments presented in this paper were conducted using the PlaFRIM experimental testbed, supported by Inria, CNRS (LABRI and IMB), Université de Bordeaux, Bordeaux INP and Conseil Régional d'Aquitaine (see https://www.plafrim.fr), as well as by the MCIA (Mésocentre de Calcul Intensif Aquitain).

This study was carried out in the framework of the University of Bordeaux's France 2030 program / RRI PHDS.

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

# Session information

```
sessionInfo()
```

```
R version 4.4.1 (2024-06-14)
Platform: x86_64-pc-linux-gnu
Running under: Ubuntu 24.04.2 LTS

Matrix products: default
BLAS:   /usr/lib/x86_64-linux-gnu/blas/libblas.so.3.12.0
LAPACK: /usr/lib/x86_64-linux-gnu/lapack/liblapack.so.3.12.0

locale:
 [1] LC_CTYPE=C.UTF-8       LC_NUMERIC=C           LC_TIME=C.UTF-8
 [4] LC_COLLATE=C.UTF-8     LC_MONETARY=C.UTF-8    LC_MESSAGES=C.UTF-8
 [7] LC_PAPER=C.UTF-8       LC_NAME=C              LC_ADDRESS=C
[10] LC_TELEPHONE=C         LC_MEASUREMENT=C.UTF-8 LC_IDENTIFICATION=C
```

```
time zone: Etc/UTC
tzcode source: system (glibc)

attached base packages:
[1] stats     graphics  grDevices datasets  utils     methods   base

other attached packages:
[1] reservoirnet_0.2.0 ggplot2_3.5.1      dplyr_1.1.4

loaded via a namespace (and not attached):
 [1] utf8_1.2.4        generics_0.1.3    tidyr_1.3.1      renv_1.0.11
 [5] rstatix_0.7.2     lattice_0.20-45   stringi_1.8.4    digest_0.6.37
 [9] magrittr_2.0.3    evaluate_1.0.1    grid_4.4.1       timechange_0.3.0
[13] fastmap_1.2.0     rprojroot_2.0.4   jsonlite_1.8.9   Matrix_1.7-1
[17] slider_0.3.2      backports_1.5.0   brio_1.1.5       Formula_1.2-5
[21] purrr_1.0.2       fansi_1.0.6       scales_1.3.0     abind_1.4-8
[25] cli_3.6.3         rlang_1.1.4       munsell_0.5.1    withr_3.0.2
[29] yaml_2.3.10       tools_4.4.1       ggsignif_0.6.4   colorspace_2.1-1
[33] ggpubr_0.6.0      here_1.0.1        broom_1.0.7      reticulate_1.40.0
[37] png_0.1-8         vctrs_0.6.5       R6_2.5.1         lifecycle_1.0.4
[41] lubridate_1.9.3   snakecase_0.11.1  stringr_1.5.1    car_3.1-3
[45] janitor_2.2.0     warp_0.2.1        pkgconfig_2.0.3  pillar_1.9.0
[49] gtable_0.3.6      Rcpp_1.0.13-1     glue_1.8.0       xfun_0.49
[53] tibble_3.2.1      tidyselect_1.2.1  knitr_1.49       farver_2.1.2
[57] htmltools_0.5.8.1 labeling_0.4.3    rmarkdown_2.29   carData_3.0-5
[61] testthat_3.2.1.1  compiler_4.4.1
```

