# OpenReview forum: "Reservoir Computing in R: a Tutorial for Using reservoirnet to Predict Complex Time-Series"
_Computo — Accepted by Computo_

### Review · Reviewer_e4tQ · 2025-03-05

**Summary Of Contributions:**

## Summary

The paper, authored by Thomas Ferté and colleagues introduces a R package for  **Reservoir Computing (RC)**, a machine learning technique that efficiently processes information generated by dynamical systems. While RC has been implemented in Python and Julia, this work adds **reservoirnet**, an R package that provides access to the **ReservoirPy**  Python wrapper (based on ScikitLearn),  allowing R users to leverage RC for time series forecasting provided that they have python installed on their machine


## Main Contributions

1. Introduction to Reservoir Computing (RC)

   - RC is based on Echo State Networks (ESNs), where only the output layer is trained while the internal recurrent connections (the reservoir) remain fixed.
   - RC efficiently captures temporal dependencies and non-linear interactions in time-series data.

2. Practical Implementation in R via `reservoirnet` from `ReservoirPy`

   - The paper provides a step-by-step tutorial on using `reservoirnet` for time-series regression and classification.
   - It showcases two examples:

     - Predicting COVID-19 hospitalizations using public health data.
     - Classifying Japanese vowels from speech datasets.

3. Advanced Case Study: Forecasting COVID-19 Hospitalizations

   - The study evaluates the performance of RC in predicting 14-day-ahead hospitalizations at Bordeaux University Hospital.
   - It compares RC to Elastic-Net Penalized Regression (Enet), previously used for COVID-19 forecasting.
   - The study investigates different RC architectures, including variations in input scaling and connections between input and output layers (skip connections ?).

4. Evaluation and Results

   - RC models are compared to Enet, with results showing that RC can be competitive in epidemic forecasting.
   - The best hyperparameters are identified using random search, tuning parameters like spectral radius, leaking rate, and ridge regularization.
   - Multiple input scalings and connections between input and output layers improve performance.
   - RC's ability to handle non-stationary processes is analyzed, showing potential for real-world hospital management applications.

**Audience:**

Yes

**Broader Impact Concerns:**

No concerns

**Claims And Evidence:**

Yes

**Requested Changes:**

##  Details


### Equations
All equations should be treated as text and be followed (most of the time) by appropriate punctuation: '','' , ''.'', '';'

### How does it relates  to  the 2 years old reservoirR github R package  ?

devtools::install_github(repo = "reservoirpy/reservoirR")


### Running the code

I was able to run all the code but had to re-install `slider` and `cli` R packages.
I did not fully understand why




## Main comments




While the authors present a comprehensive tutorial, several addition could enhance the work:

###	Performance Comparison with Advanced Models

The paper compares RC primarily with Elastic-Net Penalized Regression (Enet). Including comparisons with more advanced models, such as Long Short-Term Memory (LSTM) networks, could provide a deeper understanding of RC’s relative performance, especially given that LSTMs are widely used for time-series forecasting.

### Addressing Identified Limitations of RC

Recent studies have identified several limitations of Reservoir Computing (RC), such as sensitivity to input data changes and challenges in modeling complex dynamic systems. For instance, Zhang and Cornelius (2022) discussed the instability and regularization issues in next-generation reservoir computing.

Similarly, another study by Zhang and Cornelius (2024) highlighted how increasing data can paradoxically degrade the performance of RC models due to instability, emphasizing the need for proper regularization strategies.


To mitigate these limitations, researchers have proposed several strategies:

1.	**Enhanced Regularization Techniques:** Implementing stronger regularization methods can help stabilize RC models, especially when dealing with large datasets.

 2.	**Noise Introduction During Training:** Introducing controlled noise during the training phase can improve the generalization capabilities of RC models.

```
@article{zhang2022catch22s,
  title     = {Catch-22s of Reservoir Computing},
  author    = {Zhang, Yuanzhao and Cornelius, Sean P.},
  journal   = {arXiv preprint arXiv:2210.10211},
  year      = {2022},
  url       = {https://arxiv.org/abs/2210.10211}
}
@article{zhang2024instability,
 title   = {How More Data Can Hurt: Instability and Regularization in Next-Generation Reservoir Computing},
 author  = {Zhang, Yuanzhao and Cornelius, Sean P.},
 journal  = {arXiv preprint arXiv:2407.08641},
 year   = {2024},
 url    = {https://arxiv.org/abs/2407.08641}
}
```


### Detailed Analysis of Computational Efficiency

A more in-depth analysis of the computational efficiency of RC, especially in comparison to other models like LSTMs, would be valuable. This could include discussions on training times, resource utilization, and scalability.

### Inclusion of Visual Aids

Incorporating visualizations, such as flowcharts or diagrams, to illustrate the architecture of RC models and the workflow within the **reservoirnet** package could enhance understanding, particularly for readers new to the concept.

**Strengths And Weaknesses:**

While the paper  does not present novel scientific findings, it offers a practical tool for the R community by providing access to Reservoir Computing (RC) methodologies, particularly Echo State Networks (ESNs).

The presented **reservoirnet** R package serves as an interface (advanced wrapper) to the Python module **reservoirpy**, enabling R users to implement Reservoir Computing (RC) .

By integrating the functionalities of this Python module  into the R environment, reservoirnet enhances the accessibility of RC techniques for R users,  contributing a valuable resource to the community.

---

> ### Author Response · Authors · 2025-04-24
> **Response**
>
> Dear Reviewer e4tQ, thank you for your valuable and constructive
> feedback. We appreciate your effort in reviewing our work and have
> carefully addressed your comments.
>
> As a side note, following Reviewer 2kBF's suggestions, changes have been
> made to the `reservoirnet` package. These updates are available on the
> <https://github.com/reservoirpy/reservoirR> GitHub repository in the
> computo_response branch. Once the reviewers approve our revisions, these
> changes will be merged into the main branch and submitted to CRAN.

---

> > ### Author Response · Authors · 2025-04-24
> > **Comment 1**
> >
> > # Comment 1
> >
> > **All equations should be treated as text and be followed (most of the
> > time) by appropriate punctuation: '','' , ''.'', '';'}**
> >
> > ## Response
> >
> > We thank the reviewer for this comment and have revised the text
> > accordingly, ensuring that equations are properly punctuated.

---

> > > ### Author Response · Authors · 2025-04-24
> > > **Comment 2**
> > >
> > > # Comment 2
> > >
> > > **How does it relates to the 2 years old reservoirR github R package ?**
> > >
> > > ``` r
> > > devtools::install_github(repo = "reservoirpy/reservoirR")
> > > ```
> > >
> > > ## Response
> > >
> > > The GitHub repository reservoirpy/reseroirR is the one containing
> > > reservoirnet so they are basically the same package. We added this to
> > > the paper.
> > >
> > > ## Changes
> > >
> > > Changes were made at section 3.1 Installation.
> > >
> > > **New:** Alternatively, it can also be installed from GitHub:
> > >
> > > ```{r eval=FALSE}
> > > # Install reservoirnet package from GitHub
> > > devtools::install_github(repo = "reservoirpy/reservoirR")
> > > ```

---

> > > > ### Author Response · Authors · 2025-04-24
> > > > **Comment 3**
> > > >
> > > > # Comment 3
> > > >
> > > > **I was able to run all the code but had to re-install slider and cli R
> > > > packages. I did not fully understand why.**
> > > >
> > > > ## Response
> > > >
> > > > We apologize for the inconvenience. One possible explanation is that the
> > > > version specified in the `renv` file did not match the version
> > > > previously installed on the reviewer's system. We are happy to assist
> > > > further if needed, but we are currently unaware of the specific cause of
> > > > the issue.

---

> > > > > ### Author Response · Authors · 2025-04-24
> > > > > **Comment 4**
> > > > >
> > > > > # Comment 4
> > > > >
> > > > > **While the authors present a comprehensive tutorial, several addition
> > > > > could enhance the work:**
> > > > >
> > > > > **Performance Comparison with Advanced Models. The paper compares RC
> > > > > primarily with Elastic-Net Penalized Regression (Enet). Including
> > > > > comparisons with more advanced models, such as Long Short-Term Memory
> > > > > (LSTM) networks, could provide a deeper understanding of RC’s relative
> > > > > performance, especially given that LSTMs are widely used for time-series
> > > > > forecasting.**
> > > > >
> > > > > **Addressing Identified Limitations of RC.**
> > > > >
> > > > > **Detailed Analysis of Computational Efficiency.** **A more in-depth
> > > > > analysis of the computational efficiency of RC, especially in comparison
> > > > > to other models like LSTMs, would be valuable. This could include
> > > > > discussions on training times, resource utilization, and scalability.**
> > > > >
> > > > > ## Response
> > > > >
> > > > > We agree that the comparison in our paper is currently limited to
> > > > > Elastic-Net Penalized Regression (Enet). To provide a more comprehensive
> > > > > evaluation, we refer the reader to our previous paper, which includes an
> > > > > in-depth comparison with additional forecasting methods, such as LSTM
> > > > > networks, covering both performance and computational efficiency.
> > > > >
> > > > > ## Changes
> > > > >
> > > > > Changes were made at section 4.1 Introduction.
> > > > >
> > > > > **New:** The aim of this study is to showcase the use of `reservoirnet`
> > > > > for an advanced use case in forecasting the SARS-CoV-2 pandemic in `R`.
> > > > > Several architectural choices will be evaluated, such as the connection
> > > > > between the input layer and the output layer, and the use of either
> > > > > individual input scaling per feature or a common input scaling. The
> > > > > performance of Reservoir Computing (RC) will be compared with
> > > > > elastic-net penalized regression (identified as the optimal model in
> > > > > Ferté et al. (2022)), while a more in-depth comparison of performance
> > > > > against other methods can be found in Ferté, Dutartre, Hejblum,
> > > > > Griffier, Jouhet, Thiébaut, Legrand, et al. (2024).
> > > > >
> > > > > **Old:** *The primary aim of this study is to assess the performance of
> > > > > RC in this forecasting task. Secondary objectives include (i) comparing
> > > > > the performance of RC with that of elastic-net penalized regression
> > > > > (identified as the optimal model in Ferté et al. (2022)) and (ii)
> > > > > evaluating variations in RC performance based on different architectural
> > > > > choices, such as the connection between the input layer and the output
> > > > > layer, and the use of one input scaling per feature versus a common
> > > > > input scaling.*

---

> > > > > > ### Author Response · Authors · 2025-04-24
> > > > > > **Comment 5**
> > > > > >
> > > > > > # Comment 5
> > > > > >
> > > > > > **Recent studies have identified several limitations of Reservoir
> > > > > > Computing (RC), such as sensitivity to input data changes and challenges
> > > > > > in modeling complex dynamic systems. For instance, Zhang and Cornelius
> > > > > > (2022) discussed the instability and regularization issues in
> > > > > > next-generation reservoir computing.** **Similarly, another study by
> > > > > > Zhang and Cornelius (2024) highlighted how increasing data can
> > > > > > paradoxically degrade the performance of RC models due to instability,
> > > > > > emphasizing the need for proper regularization strategies.** **To
> > > > > > mitigate these limitations, researchers have proposed several
> > > > > > strategies: i) Enhanced Regularization Techniques: Implementing stronger
> > > > > > regularization methods can help stabilize RC models, especially when
> > > > > > dealing with large datasets ; ii) Noise Introduction During Training:
> > > > > > Introducing controlled noise during the training phase can improve the
> > > > > > generalization capabilities of RC models.**
> > > > > >
> > > > > > ``` r
> > > > > > @article{zhang2022catch22s,
> > > > > >   title     = {Catch-22s of Reservoir Computing},
> > > > > >   author    = {Zhang, Yuanzhao and Cornelius, Sean P.},
> > > > > >   journal   = {arXiv preprint arXiv:2210.10211},
> > > > > >   year      = {2022},
> > > > > >   url       = {https://arxiv.org/abs/2210.10211}
> > > > > > }
> > > > > > @article{zhang2024instability,
> > > > > >  title    = {How More Data Can Hurt: Instability and Regularization
> > > > > >              in Next-Generation Reservoir Computing},
> > > > > >  author   = {Zhang, Yuanzhao and Cornelius, Sean P.},
> > > > > >  journal  = {arXiv preprint arXiv:2407.08641},
> > > > > >  year     = {2024},
> > > > > >  url      = {https://arxiv.org/abs/2407.08641}
> > > > > > }
> > > > > > ```
> > > > > >
> > > > > > ## Response
> > > > > >
> > > > > > Thank you for recommending the two very interesting papers. We agree
> > > > > > that it is important to highlight the limitations, challenges, and
> > > > > > future opportunities of reservoir computing. To this end, we have added
> > > > > > the recent article by Yan et al. (Yan, M., Huang, C., Bienstman, P. et
> > > > > > al. Emerging opportunities and challenges for the future of reservoir
> > > > > > computing. Nat Commun 15, 2056 (2024).
> > > > > > https://doi.org/10.1038/s41467-024-45187-1) to the introduction, as it
> > > > > > specifically addresses these aspects in a comprehensive manner.
> > > > > >
> > > > > > While the proposed articles are indeed insightful, they focus on
> > > > > > particular challenges within the field. Given that our manuscript is
> > > > > > primarily intended as a tutorial, we felt that including a more general
> > > > > > peer-reviewed article would be more aligned with our goals. That said,
> > > > > > we are happy to reconsider this point and would welcome further
> > > > > > suggestions or guidance from the reviewer or the editor.
> > > > > >
> > > > > > ## Changes
> > > > > >
> > > > > > Changes were made at section 1 Introduction.
> > > > > >
> > > > > > **New:** This provides interesting and potentially more efficient
> > > > > > alternative to traditional machine learning computing and might play an
> > > > > > important role in the coming years (Yan et al. 2024).
> > > > > >
> > > > > > **Old:** *This provides interesting and potentially more efficient
> > > > > > alternative to traditional machine learning computing.*

---

> > > > > > > ### Author Response · Authors · 2025-04-24
> > > > > > > **Comment 6**
> > > > > > >
> > > > > > > # Comment 6
> > > > > > >
> > > > > > > **Incorporating visualizations, such as flowcharts or diagrams, to
> > > > > > > illustrate the architecture of RC models and the workflow within the
> > > > > > > `reservoirnet` package could enhance understanding, particularly for
> > > > > > > readers new to the concept.**
> > > > > > >
> > > > > > > ## Response
> > > > > > >
> > > > > > > In addition to Figure 2, which illustrates the package's workflow, we
> > > > > > > have added two new figures—one for each basic use case—to visually
> > > > > > > represent the reservoir model built for each scenario.
> > > > > > >
> > > > > > > ## Changes:
> > > > > > >
> > > > > > > We added a figure to the Basic Regression Use Case (figure 4) section
> > > > > > > along with the following text:
> > > > > > >
> > > > > > > **New:** The objective of this task is to train a RC model using the
> > > > > > > input features to forecast the number of hospitalized patients 14 days
> > > > > > > ahead, as illustrated in Figure 4.
> > > > > > >
> > > > > > > We also added a figure to the Classification Use Case section (figure
> > > > > > > 10) along with the following text:
> > > > > > >
> > > > > > > Figure 10 illustrates this task.

---

> > > > > > > > ### Comment · Reviewer_e4tQ · 2025-05-12
> > > > > > > > **Response to review**
> > > > > > > >
> > > > > > > > I believe the responses have contributed to improving the quality of the paper. The remaining question I have, which is more fundamental, is: does a Journal of Statistical Software-type paper fall within the scope of Computo? If so, I am clearly in favor of accepting the paper; otherwise, it’s something that should be discussed.

---

> > > > > > > ### Author Response · Authors · 2025-04-24
> > > > > > > **Remark to Editors on comment 4 of reviewer e4tQ**
> > > > > > >
> > > > > > > We would like to bring to your attention our response to Reviewer e4tQ.
> > > > > > > The reviewer suggests citing two non-peer-reviewed papers authored by
> > > > > > > the same team, which we believe are not fully relevant to the scope and
> > > > > > > purpose of our tutorial article :
> > > > > > >
> > > > > > > ``` r
> > > > > > > @article{zhang2022catch22s,
> > > > > > >   title     = {Catch-22s of Reservoir Computing},
> > > > > > >   author    = {Zhang, Yuanzhao and Cornelius, Sean P.},
> > > > > > >   journal   = {arXiv preprint arXiv:2210.10211},
> > > > > > >   year      = {2022},
> > > > > > >   url       = {https://arxiv.org/abs/2210.10211}
> > > > > > > }
> > > > > > > @article{zhang2024instability,
> > > > > > >  title    = {How More Data Can Hurt: Instability and Regularization
> > > > > > >              in Next-Generation Reservoir Computing},
> > > > > > >  author   = {Zhang, Yuanzhao and Cornelius, Sean P.},
> > > > > > >  journal  = {arXiv preprint arXiv:2407.08641},
> > > > > > >  year     = {2024},
> > > > > > >  url      = {https://arxiv.org/abs/2407.08641}
> > > > > > > }
> > > > > > > ```
> > > > > > >
> > > > > > > We have instead included a more general and comprehensive published
> > > > > > > reference in the introduction:
> > > > > > >
> > > > > > > ``` r
> > > > > > > @article{yan2024emerging,
> > > > > > >   title={Emerging opportunities and challenges for the future of reservoir computing},
> > > > > > >   author={Yan, Min and Huang, Can and Bienstman, Peter and Tino, Peter and Lin, Wei and Sun, Jie},
> > > > > > >   journal={Nature Communications},
> > > > > > >   volume={15},
> > > > > > >   number={1},
> > > > > > >   pages={2056},
> > > > > > >   year={2024},
> > > > > > >   publisher={Nature Publishing Group UK London}
> > > > > > > }
> > > > > > > ```
> > > > > > >
> > > > > > > We remain open to further guidance on this matter.

---

### Review · Reviewer_LM72 · 2025-03-09

**Summary Of Contributions:**

This work introduces an R interface to the tools of 'reservoirpy', a Python module implementing the statistical approach known as Reservoir Computing (RC). RC is a machine learning method based on neural networks, where a structured core enables a transition between inputs and outputs via random mixing steps. This allows for the modeling of complex dynamic phenomena (such as chronological non-stationarity or non-linearity) in a more computationally efficient manner than LSTMs, which are typically used in this context. While this is not original research, it provides a valuable practical resource for the R community.

**Audience:**

Yes

**Broader Impact Concerns:**

/

**Claims And Evidence:**

Yes

**Requested Changes:**

What I mentioned in the previous paragraph could lead to some modifications, if the authors agree.

By way of minor comments, I must add that the writing includes some awkward formulations, of which I give a few examples below:
- formula (1): what is tanh of a vector?
- p.4: the spectral radius is the largest eigenvalue 'in modulus'
- p.7: 'subsequently'... 'subsequent'
- fig. 3: number 'ofx' RT-PCR
- p. 16: while the primary objective is clearly understandable, I find it difficult to grasp the purpose of the secondary one, perhaps some additional explanations would be helpful
- p. 23: 'compartmental'
- p. 24-25: the expressions of MAE, MRE, MAEB and MREB could be presented in a more mathematical manner
- p. 25: 'with on'
- p. 26: 'other hyparameter' and  'do not show'
- fig. 15: could we use the numerous forecasts to get some confidence intervals?

**Strengths And Weaknesses:**

The step-by-stel tutorial is precise and very helpful for users who are not necessarily experts. Two examples are widely discussed (prediction of covid-19 hospitalizations and classification of Japanese vowels). RC is essentially compared with Elastic-Net, from various architectures (input scaling, inputs connected to outputs, tuning of hyperparameters, etc.)

Nevertheless, a question arises: do the authors simply aim to explain the use of their new package (in which case the comparative study of RC might not need to be as detailed, since it is not the main objective), or do they also intend to analyze the performances of RC (in which case other predictive models besides Elastic-Net should be considered for comparison)? This point could be clarified.

I encountered many technical difficulties in getting the connection with 'reservoirpy'. A paragraph on the prerequisites related to the Python distribution and its modules would be really valuable for non-experts. In addition, the R package 'slider' is required, although it is not provided in the example code.

I also have a technical question: page 4, $W_{in}$ and $W$ are presented as 'sparse matrices' (i.e. with lots of 0) and in the same paragraph as taking binomial and gaussian values, respectively. How is this possible? This mathematical part has to be corrected or at least clarified.

---

> ### Author Response · Authors · 2025-04-24
> **Response**
>
> Dear Reviewer LM72, thank you for your valuable and constructive
> feedback. We appreciate your effort in reviewing our work and have
> carefully addressed your comments.
>
> As a side note, following Reviewer 2kBF's suggestions, changes have been
> made to the `reservoirnet` package. These updates are available on the
> <https://github.com/reservoirpy/reservoirR> GitHub repository in the
> computo_response branch. Once the reviewers approve our revisions, these
> changes will be merged into the main branch and submitted to CRAN.

---

> > ### Author Response · Authors · 2025-04-24
> > **Comment 1**
> >
> > # Comment 1
> >
> > **Nevertheless, a question arises: do the authors simply aim to explain
> > the use of their new package (in which case the comparative study of RC
> > might not need to be as detailed, since it is not the main objective),
> > or do they also intend to analyze the performances of RC (in which case
> > other predictive models besides Elastic-Net should be considered for
> > comparison)? This point could be clarified.**
> >
> > ## Response
> >
> > The primary purpose of this paper is to demonstrate the use of the
> > `reservoirnet` package. We believed that showcasing its application on a
> > real-world complex problem would help readers appreciate its potential
> > and ease of use. We agree that the claim about the objective of the
> > final section should be adjusted accordingly. Additionally, we refer the
> > reader to published, more in-depth, comparisons of methods.
> >
> > That being said, if the reviewer still feels that this section is too
> > detailed, we are open to removing the elastic-net comparison to simplify
> > the paper or exploring other potential simplifications.
> >
> > ## Changes
> >
> > We changed the followings in section 4.1 Introduction.
> >
> > **New:** The aim of this study is to showcase the use of `reservoirnet`
> > for an advanced use case in forecasting the SARS-CoV-2 pandemic in `R`.
> > Several architectural choices will be evaluated, such as the connection
> > between the input layer and the output layer, and the use of either
> > individual input scaling per feature or a common input scaling. The
> > performance of Reservoir Computing (RC) will be compared with
> > elastic-net penalized regression (identified as the optimal model in
> > Ferté et al. (2022)), while a more in-depth comparison of performance
> > against other methods can be found in Ferté, Dutartre, Hejblum,
> > Griffier, Jouhet, Thiébaut, Legrand, et al. (2024).
> >
> > **Old:** *The primary aim of this study is to assess the performance of
> > RC in this forecasting task. Secondary objectives include (i) comparing
> > the performance of RC with that of elastic-net penalized regression
> > (identified as the optimal model in Ferté et al. (2022)) and (ii)
> > evaluating variations in RC performance based on different architectural
> > choices, such as the connection between the input layer and the output
> > layer, and the use of one input scaling per feature versus a common
> > input scaling.*

---

> > > ### Author Response · Authors · 2025-04-24
> > > **Comment 2**
> > >
> > > # Comment 2
> > >
> > > **I encountered many technical difficulties in getting the connection
> > > with `reservoirpy`. A paragraph on the prerequisites related to the
> > > Python distribution and its modules would be really valuable for
> > > non-experts.**
> > >
> > > ## Response
> > >
> > > We agree that the installation process should be as smooth as possible.
> > > To facilitate this, we have introduced the `install\_reservoirpy()`
> > > function, which automatically installs the `reservoirpy` module. For
> > > this function to work, the user will need `Python` version 3.8 or
> > > higher.
> > >
> > > ## Changes
> > >
> > > We added the following at section 3.1 Installation:
> > >
> > > **New:** For `reservoirnet` to work, it will require Python version 3.8
> > > or higher, along with the `reservoirpy` module which can be installed
> > > with the `install_reservoirpy()` function:
> > >
> > > ```{r eval=FALSE}
> > > reservoirnet::install_reservoirpy()
> > > ```

---

> > > > ### Author Response · Authors · 2025-04-24
> > > > **Comment 3**
> > > >
> > > > # Comment 3
> > > >
> > > > **In addition, the `R` package `slider` is required, although it is not
> > > > provided in the example code.**
> > > >
> > > > ## Response
> > > >
> > > > The `slider` package is used in Section 3.3.1 (Covid-19 Data) in the
> > > > second code chunk to compute the derivatives. We are open to providing
> > > > further clarification or making modifications to improve the
> > > > presentation, should the reviewer feel it necessary.
> > > >
> > > > ## Changes
> > > >
> > > > No change made.

---

> > > > > ### Author Response · Authors · 2025-04-24
> > > > > **Comment 4**
> > > > >
> > > > > # Comment 4
> > > > >
> > > > > **I also have a technical question: page 4, Win and W are presented as
> > > > > 'sparse matrices' (i.e. with lots of 0) and in the same paragraph as
> > > > > taking binomial and gaussian values, respectively. How is this possible?
> > > > > This mathematical part has to be corrected or at least clarified.**
> > > > >
> > > > > ## Response
> > > > >
> > > > > We revised this section to provide a clearer explanation of how the
> > > > > sparse matrices are defined. The process involves three steps: first,
> > > > > dense matrices are generated — using a binomial distribution for
> > > > > $W_{in}$ and a Gaussian distribution for $W$. Next, a sparsity mask is
> > > > > applied to impose the desired sparsity. Finally, for $W$ only, the
> > > > > matrix is scaled to match the specified spectral radius.
> > > > >
> > > > > ## Changes
> > > > >
> > > > > We made the following changes at section 2 RC presentation:
> > > > >
> > > > > **New:** The input-reservoir connection matrix ($W_{in}$) and the
> > > > > intra-reservoir connection matrix ($W$) are generated in three steps.
> > > > > $W_{in}$ is generated using a Bernoulli (bimodal) distribution where
> > > > > each value can be either $-I_{scale}(m)$ or $I_{scale}(m)$ with an equal
> > > > > probability where $m = 1, \dots, M$ corresponds to a given feature in
> > > > > the input layer. The input scaling, denoted $I_{scale}$, is a
> > > > > hyperparameter coefficient common to all features from the input layer
> > > > > or specific to each feature $m$. In that case, the more important the
> > > > > feature is, the greater should be its input scaling. $W$ is generated
> > > > > from a Gaussian distribution $\mathcal{N}(0,1)$. Both $W_{in}$ and $W$
> > > > > then undergo sparsification, where a connectivity mask is applied to
> > > > > retain only $10\%$ of the connections, enforcing sparsity. In a third
> > > > > step, the $W$ matrix is scaled according to the defined spectral radius,
> > > > > a hyperparameter defining the highest eigen value of $W$.
> > > > >
> > > > > **Old:** $W_{in}$ is a matrix (usually sparse) generated using a
> > > > > Bernoulli (bimodal) distribution where each value can be either
> > > > > $-I_{scale}(m)$ or $I_{scale}(m)$ with an equal probability where
> > > > > $m = 1, \dots, M$ corresponds to a given feature in the input layer. The
> > > > > input scaling, denoted $I_{scale}$, is a hyperparameter coefficient
> > > > > common to all features from the input layer or specific to each feature
> > > > > $m$. In that case, the more important the feature is, the greater should
> > > > > be its input scaling. $W$ is a matrix (usually sparse) where values are
> > > > > generated from a Gaussian distribution $\mathcal{N}(0,1)$. Then, the $W$
> > > > > matrix is scaled according to the defined spectral radius, a
> > > > > hyperparameter defining the highest eigen value of $W$.

---

> > > > > > ### Author Response · Authors · 2025-04-24
> > > > > > **Comment 5**
> > > > > >
> > > > > > # Comment 5
> > > > > >
> > > > > > **By way of minor comments, I must add that the writing includes some
> > > > > > awkward formulations, of which I give a few examples below:**
> > > > > >
> > > > > > -   **formula (1): what is tanh of a vector?**
> > > > > > -   **p.4: the spectral radius is the largest eigenvalue 'in modulus'**
> > > > > > -   **p.7: 'subsequently'... 'subsequent'**
> > > > > > -   **fig. 3: number 'ofx' RT-PCR**
> > > > > > -   **p. 16: while the primary objective is clearly understandable, I
> > > > > >     find it difficult to grasp the purpose of the secondary one, perhaps
> > > > > >     some additional explanations would be helpful**
> > > > > > -   **p. 23: 'compartmental'**
> > > > > > -   **p. 24-25: the expressions of MAE, MRE, MAEB and MREB could be
> > > > > >     presented in a more mathematical manner**
> > > > > > -   **p. 25: 'with on'**
> > > > > > -   **p. 26: 'other hyparameter' and 'do not show'**
> > > > > > -   **fig. 15: could we use the numerous forecasts to get some
> > > > > >     confidence intervals?**
> > > > > >
> > > > > > ## Response
> > > > > >
> > > > > > We have modified the text accordingly and corrected other grammatical
> > > > > > mistakes throughout the manuscript. Regarding the comment on obtaining
> > > > > > confidence intervals from the numerous forecasts, to our knowledge,
> > > > > > there is currently no direct method for deriving confidence intervals
> > > > > > from the aggregation of multiple Reservoir Computing approaches.
> > > > > > Alternatives, such as conformal prediction, could be considered, but
> > > > > > this falls outside the scope of the present work.
> > > > > >
> > > > > > ## Changes
> > > > > >
> > > > > > **New:**
> > > > > >
> > > > > > -   The function $tanh()$ represents the activation function, applied
> > > > > >     element-wise to each component of the vector, ensuring that each
> > > > > >     node’s activation is scaled between $-1$ and $1$.
> > > > > > -   Spectral radius (`sr`): the spectral radius is the largest
> > > > > >     eigenvalue in modulus of the reservoir connectivity matrix.
> > > > > > -   To accomplish this, we will initially train our model on data
> > > > > >     preceding the date of January 1, 2022, and then apply it to forecast
> > > > > >     values using the following dataset.
> > > > > > -   number of RT-PCR
> > > > > > -   While this second approach may seem somewhat superfluous in this
> > > > > >     context, it could be useful, for example, in cases where multiple
> > > > > >     speakers take turns speaking, allowing us to identify which sequence
> > > > > >     belongs to each individual speaker.
> > > > > > -   'compartmental' was replaced by 'mechanistic'. Those involve
> > > > > >     differential equation to model the epidemic.
> > > > > > -   Metrics were defined as follow:
> > > > > >     -   the mean absolute error:
> > > > > >         $\text{MAE} = \frac{1}{T} \sum_{t=1}^{T} \left| \hat{hosp}_{t+14} - hosp_{t+14} \right|$.
> > > > > >     -   the median relative error:
> > > > > >         $\text{MRE} = \text{median} \left( \left| \frac{\hat{hosp}_{t+14} - hosp_{t+14}}{hosp_{t+14}} \right| \right)$.
> > > > > >     -   the mean absolute error to baseline:
> > > > > >         $\text{MAEB} = \frac{1}{T} \sum_{t=1}^{T} \left( \left| \hat{hosp}_{t+14} - hosp_{t+14} \right| - \left| hosp_{t} - hosp_{t+14} \right| \right)$.
> > > > > >     -   the median relative error to baseline:
> > > > > >         $\text{MREB} = \text{median} \left( \left| \frac{\hat{hosp}_{t+14} - hosp_{t+14}}{hosp_{t} - hosp_{t+14}} \right| \right)$)
> > > > > > -   First we compared RC with a single input scaling common to all
> > > > > >     features and a RC with a specific input scaling per feature.
> > > > > > -   The remaining features do not exhibit a clear pattern.
> > > > > >
> > > > > > **Old:**
> > > > > >
> > > > > > -   *..*
> > > > > > -   *Spectral radius (`sr`): the spectral radius is the maximum absolute
> > > > > >     eigenvalue of the reservoir connectivity matrix.*
> > > > > > -   *To accomplish this, we will initially train our model on data
> > > > > >     preceding the date of January 1, 2022, and subsequently apply it to
> > > > > >     forecast values using the subsequent dataset.*
> > > > > > -   *number 'ofx' RT-PCR*
> > > > > > -   *..*
> > > > > > -   *'compartmental'*
> > > > > > -
> > > > > >     -   *the mean absolute error:*
> > > > > >         $MAE = mean(|\widehat{hosp_{t+14}}-hosp_{t+14}|)$*.*
> > > > > >     -   *the median relative error:*
> > > > > >         $MRE = median(|\frac{\widehat{hosp_{t+14}}-hosp_{t+14}}{hosp_{t+14}}|)$*.*
> > > > > >     -   *the mean absolute error to baseline:*
> > > > > >         $MAEB =  mean(|\widehat{hosp_{t+14}}-hosp_{t+14}| - |hosp_{t}-hosp_{t+14}|)$*.*
> > > > > >     -   *the median relative error to baseline:*
> > > > > >         $MREB = median(|\frac{\widehat{hosp_{t+14}}-hosp_{t+14}}{hosp_{t}-hosp_{t+14}}|)$*)*
> > > > > > -   *First we compared RC with a single input scaling common to all
> > > > > >     features and a RC with on specific input scaling per feature.*
> > > > > > -   *The rest of the feature do not show clear pattern.*

---

### Review · Reviewer_2kBF · 2025-03-11

**Summary Of Contributions:**

This paper presents `reservoirnet`, an R interface for 'reservoirpy', a Python toolkit that leverages Reservoir Computing (RC) for machine learning tasks. The paper introduces the main concepts behind reservoir computing and then proceeds with a list of tutorials of increasing complexity to highlight the benefits of the package for R users.

**Audience:**

Yes

**Broader Impact Concerns:**

-

**Claims And Evidence:**

Yes

**Requested Changes:**

* Update code to make things simpler when possible (cf comment above)
* If your goal is to evaluate the performance of RC at the considered task, include more reasonable baselines, if it is rather to showcase your R module, you should probably edit your claim

**Strengths And Weaknesses:**

I find the paper rather useful, though it seems that sometimes, the authors could make the examples simpler to better help the reader grasp the important typical usages of the library and their implementation.

Typically, I don't understand why sometimes you pass a model to your fit and sometimes you just pass the readout, to me it seems that this paper is a tutorial to help beginners get their hands on this R package, and in this case, it would be nice to make your examples as simple as possible, and as coherent as possible.

```R
model <- reservoirnet::link(reservoir, readout)

[...]

fit <- reservoirnet::reservoirR_fit(node = model,
                                    X = lsdf$xTrain,
                                    Y = lsdf$yTrain,
                                    warmup = 30,
                                    reset = TRUE)
```

Then, for the sequence-to-vector case, you have:

```R
res <- reservoirnet::reservoirR_fit(readout,X = states_train, Y = Y_train)
```

because you want to rely on the last state only.
Shouldn't it be implemented in the library itself, as a parameter to the `reservoirR_fit` function, to reduce the burden for the end user?

Then, the next model is defined as:

```R
source <- createNode("Input")
readout <- createNode("Ridge",ridge=1e-6)
reservoir <- createNode("Reservoir",units = 500,lr=0.1, sr=0.9, seed = 1)
# Connect the input layer to the reservoir and connect both the input layer and
# the reservoir to the output layer
model <- list(source %>>% reservoir, source) %>>% readout
```

Is the goal here to introduce a new way to build a model? If so, this should be discussed, otherwise, you should use a uniform way to build your models.
Also, it is unclear to me why in this case you want to connect both the input layer and the reservoir to the output layer, and not just the reservoir.

Also, for the last application (prediction of the COVID19 load at hospitals), you claim that "The primary aim of this study is to assess the performance of RC in this forecasting task."
If so, the considered models should be experimentally compared to solid forecasting baselines.
If the goal is rather to show that RC is able to handle such tasks, then the initial claim should be lowered.

---

> ### Author Response · Authors · 2025-04-24
> **Response**
>
> Dear Reviewer 2kBF, thank you for your valuable and constructive
> feedback. We appreciate your effort in reviewing our work and have
> carefully addressed your comments. Notably, we have made changes to the
> `reservoirnet` package following your suggestions. These updates are
> available on the <https://github.com/reservoirpy/reservoirR> GitHub
> repository in the computo_response branch. Once the reviewers approve
> our revisions, these changes will be merged into the main branch and
> submitted to CRAN.

---

> > ### Author Response · Authors · 2025-04-24
> > **Comment 1**
> >
> > # Comment 1
> >
> > ## Comment
> >
> > **Typically, I don't understand why sometimes you pass a model to your
> > fit and sometimes you just pass the readout, to me it seems that this
> > paper is a tutorial to help beginners get their hands on this R package,
> > and in this case, it would be nice to make your examples as simple as
> > possible, and as coherent as possible.**
> >
> > ``` r
> > model <- reservoirnet::link(reservoir, readout)
> >
> > [...]
> >
> > fit <- reservoirnet::reservoirR_fit(node = model,
> >                                     X = lsdf$xTrain,
> >                                     Y = lsdf$yTrain,
> >                                     warmup = 30,
> >                                     reset = TRUE)
> > ```
> >
> > **Then, for the sequence-to-vector case, you have:**
> >
> > ``` r
> > res <- reservoirnet::reservoirR_fit(readout,X = states_train, Y = Y_train)
> > ```
> >
> > **because you want to rely on the last state only. Shouldn't it be
> > implemented in the library itself, as a parameter to the
> > `reservoirR_fit` function, to reduce the burden for the end user?**
> >
> > ## Response
> >
> > Indeed, easing and simplifying this process would improve usability.
> > While this functionality would ideally be integrated into the Python
> > `reservoirpy` module itself, we have introduced a new function in the R
> > package `reservoirnet` in the meantime: `last_reservoir_state()`. This
> > function automates the extraction of the final reservoir state,
> > streamlining data preparation for sequence-to-vector tasks and ensuring
> > a smoother user experience.
> >
> > Changes to the package have been added to the GitHub repository
> > <https://github.com/reservoirpy/reservoirR> in the `computo_response`
> > branch. These updates will be merged into the main branch and
> > subsequently submitted to CRAN once the reviewers approve our revisions.
> >
> > ## Changes
> >
> > We changed the following in section 3.4.2 Classification
> > (sequence-to-vector model):
> >
> > **New:** In the context of the sequence-to-vector encoding task, only
> > the final state is used. To simplify this process, we introduce the
> > `last_reservoir_state()` function, which extracts the final reservoir
> > state. This process is executed as follows:
> >
> > ``` r
> > states_train <- reservoirnet::last_reservoir_state(node = reservoir, X = X_train)
> >
> > ...
> >
> > res <- reservoirnet::reservoirR_fit(node = readout, X = states_train, Y = Y_train)
> > ```
> >
> > Then, we use only the final state for prediction. We first extract the
> > final state using the `last_reservoir_state()` function and then use the
> > trained readout to predict the vowel using the `predict_seq()` function
> > with the `seq_to_vec` parameter set to `TRUE`:
> >
> > ``` r
> > states_test <- reservoirnet::last_reservoir_state(node = reservoir, X = X_test)
> > Y_pred <- reservoirnet::predict_seq(node = readout, X = states_test, seq_to_vec = TRUE)
> > ```
> >
> > **Old:**
> >
> > *In the context of the sequence-to-vector encoding task, only this
> > ultimate state is employed. This process is executed as follows:*
> >
> > ``` r
> > states_train = list()
> > k <- 1
> > for (x in X_train) {
> >   # Loop over each training sample in X_train.
> >   states <- reservoirnet::predict_seq(node = reservoir, X = x,
> >                                       reset=TRUE)
> >   # Generate the reservoir states for the current training sample using the
> >   # 'predict_seq' function.
> >   states_train[[k]] <- t(as.matrix(states[nrow(states),]))
> >   # Extract the final reservoir state for the current sample and store it in
> >   # 'states_train'.
> >   k <- k+1
> > }
> >
> > ...
> >
> > res <- reservoirnet::reservoirR_fit(readout,X = states_train, Y = Y_train)
> > ```
> >
> > *The prediction is also modified using only the final state:*
> >
> > ``` r
> > # The operation is repeated for the test set:
> > # - the last state vector is extracted for each utterance
> > # - the model is used to forecast the utterance from this last state vector
> > Y_pred <- list()
> > k <- 1
> > for (x in X_test) {
> >   states <- reservoirnet::predict_seq(node = reservoir, X = x,
> >                                       reset=TRUE)
> >   y <- reservoirnet::predict_seq(node = readout,
> >                                  X = as.array(states[nrow(states),]))
> >   Y_pred[[k]] <- y
> >   k <- k+1
> > }
> > ```

---

> > > ### Author Response · Authors · 2025-04-24
> > > **Comment 2**
> > >
> > > # Comment 2
> > >
> > > **Then, the next model is defined as:**
> > >
> > > ``` r
> > > source <- createNode("Input")
> > > readout <- createNode("Ridge",ridge=1e-6)
> > > reservoir <- createNode("Reservoir",units = 500,lr=0.1, sr=0.9, seed = 1)
> > > # Connect the input layer to the reservoir and connect both the input layer and
> > > # the reservoir to the output layer
> > > model <- list(source %>>% reservoir, source) %>>% readout
> > > ```
> > >
> > > **Is the goal here to introduce a new way to build a model? If so, this
> > > should be discussed, otherwise, you should use a uniform way to build
> > > your models.**
> > >
> > > ## Response
> > >
> > > The tutorial should indeed be simplified, to ensure clarity. The
> > > `link()` function and the `%>>%` operator both perform the same
> > > operation, which is to connect nodes in the model. The purpose of using
> > > both was to demonstrate different ways to achieve this. However, to
> > > simplify the tutorial for new users, we have decided to use only the
> > > `%>>%` operator throughout the manuscript and remove the `link()`
> > > function. This provides a uniform approach to building models and helps
> > > avoid any confusion.
> > >
> > > ## Changes
> > >
> > > We changed the following at section 3.3.3.1 Set the ESN:
> > >
> > > **New:**
> > >
> > > We connect the output layer to the reservoir, with the `%>>%` operator,
> > > making the model ready to be trained.
> > >
> > > ...
> > >
> > > ``` r
> > > model <- reservoir %>>% readout
> > > # Link the reservoir and readout nodes to form a complete reservoir computing
> > > # model. The '\%>>\%' operator connects the high-dimensional state generated by
> > > ```
> > >
> > > **Old:**
> > >
> > > *We connect the output layer to the reservoir, with the `link()`
> > > function, making the model ready to be trained.*
> > >
> > > *...*
> > >
> > > ``` r
> > > model <- reservoirnet::link(reservoir, readout)
> > > # Link the reservoir and readout nodes to form a complete reservoir computing
> > > # model. The 'link' function connects the high-dimensional state generated by
> > > ```

---

> > > > ### Author Response · Authors · 2025-04-24
> > > > **Comment 3**
> > > >
> > > > # Comment 3
> > > >
> > > > **Also, it is unclear to me why in this case you want to connect both the input layer and the reservoir to the output layer, and not just the reservoir.**
> > > >
> > > > ## Response
> > > >
> > > > Connecting both the input layer and the reservoir to the output layer
> > > > can be beneficial when the relationship between the input sequences and
> > > > the output is partially linear. For instance, in the case of the
> > > > SARS-CoV-2 epidemics, as shown in Table 1 of Section 4.3.2, models that
> > > > connect both the input layer and the reservoir to the output tend to
> > > > outperform models that only connect the reservoir to the output.
> > > >
> > > > From a tutorial perspective, we included this architecture to illustrate
> > > > two different ways to structure models for users. However, we understand
> > > > that in the case of vowel recognition, it may not be entirely clear
> > > > whether this linear connection is necessary. A formal comparison would
> > > > indeed be valuable, but we have already conducted such comparisons in
> > > > the advanced use case in Section 4. Given that, we believe performing
> > > > this comparison systematically across all use cases might overwhelm the
> > > > reader, so we chose not to include it here.
> > > >
> > > > ## Changes
> > > >
> > > > We changed at section 3.4.3 Transduction (sequence-to-sequence model):
> > > >
> > > > **New:** Then we can train a simple Echo State Network to solve this
> > > > task. For this example, we will connect both the input layer and the
> > > > reservoir layer to the readout layer, which is performed by the `%>>%`
> > > > operator. This direct connection between the input layer and the output
> > > > layer can be particularly useful when the relationship between the input
> > > > sequences and the output is partially linear, potentially improving
> > > > performance. Section 4 will explore this aspect in more detail through
> > > > the SARS-CoV-2 prediction task.
> > > >
> > > > **Old:** *Then we can train a simple Echo State Network to solve this
> > > > task. For this example we will connect both the input layer and the
> > > > reservoir layer to the readout layer which is performed by the `%>>%`
> > > > operator:*

---

> > > > > ### Author Response · Authors · 2025-04-24
> > > > > **Comment 4**
> > > > >
> > > > > # Comment 4
> > > > >
> > > > > **Also, for the last application (prediction of the COVID19 load at
> > > > > hospitals), you claim that "The primary aim of this study is to assess
> > > > > the performance of RC in this forecasting task." If so, the considered
> > > > > models should be experimentally compared to solid forecasting baselines.
> > > > > If the goal is rather to show that RC is able to handle such tasks, then
> > > > > the initial claim should be lowered.**
> > > > >
> > > > > **If your goal is to evaluate the performance of RC at the considered
> > > > > task, include more reasonable baselines, if it is rather to showcase
> > > > > your R module, you should probably edit your claim**
> > > > >
> > > > > ## Response
> > > > >
> > > > > We agree with the reviewer that our claim should be adjusted since our
> > > > > comparison is limited to elastic-net. To provide a more comprehensive
> > > > > evaluation, we refer the reader to our previous paper, which includes an
> > > > > in-depth comparison with additional forecasting methods.
> > > > >
> > > > > ## Changes
> > > > >
> > > > > We changed the followings at section 4.1 Introduction.
> > > > >
> > > > > **New:** The aim of this study is to showcase the use of `reservoirnet`
> > > > > for an advanced use case in forecasting the SARS-CoV-2 pandemic in `R`.
> > > > > Several architectural choices will be evaluated, such as the connection
> > > > > between the input layer and the output layer, and the use of either
> > > > > individual input scaling per feature or a common input scaling. The
> > > > > performance of Reservoir Computing (RC) will be compared with
> > > > > elastic-net penalized regression (identified as the optimal model in
> > > > > Ferté et al. (2022)), while a more in-depth comparison of performance
> > > > > against other methods can be found in Ferté, Dutartre, Hejblum,
> > > > > Griffier, Jouhet, Thiébaut, Legrand, et al. (2024).
> > > > >
> > > > > **Old:** *The primary aim of this study is to assess the performance of
> > > > > RC in this forecasting task. Secondary objectives include (i) comparing
> > > > > the performance of RC with that of elastic-net penalized regression
> > > > > (identified as the optimal model in Ferté et al. (2022)) and (ii)
> > > > > evaluating variations in RC performance based on different architectural
> > > > > choices, such as the connection between the input layer and the output
> > > > > layer, and the use of one input scaling per feature versus a common
> > > > > input scaling.*

---

### Comment · Editors_In_Chief · 2025-01-09
**Actions before sending for review**

Dear authors,

Some fixes prior to reviewing

[ ] Please add link to the gh-page in your repository description/about (https://thomasferte.github.io/reservoirnet_computo/)
[ ] Double check that you set up renv with the minimal many dependencies

Thanks a lot for your submission 👍

jc

---

> ### Author Response · Authors · 2025-01-09
> **response to: Actions before sending for review**
>
> Dear Editors,
>
> Thank you for your comments and suggestions. We have carefully addressed them and made the following modifications:
>
> Please add link to the gh-page in your repository description/about (https://thomasferte.github.io/reservoirnet_computo/)
>
> - We have added the sentence, "The draft is available as a GitHub page at https://thomasferte.github.io/reservoirnet_computo/," to the README file in the GitHub repository.
>
> Double check that you set up renv with the minimal many dependencies
>
> - We reviewed and optimized the project dependencies, removing 13 unnecessary R packages.
>
> We appreciate your feedback and look forward to the next steps in the review process.
>
> Best regards,
> Thomas Ferté

---

### Comment · Action_Editor_bUhc · 2025-03-10
**About the reviewing process**

Dear all,

Contrary to the messages sent by the OpenReview system, we are awaiting the report of a third reviewer. Authors can already take note of the comments of the first two reviewers, but in any case, the rebuttal period will be postponed until the last report has been submitted. An editorial decision will be taken at the end of this period, during which authors will have the opportunity to justify themselves and make significant changes to their submission.

Sorry for the contradictory signals sent by the platform and my own (I control the latter, but not completely the former).

jc

---

### Comment · Action_Editor_bUhc · 2025-03-12
**Rebuttal period**

Dear authors,

We now have received 3 reports that raise pertinent points regarding your submission.

A rebuttal period of 6 weeks (unit April 21st) is allowed during which you can make any changes to your submission that you feel are necessary. At the end of this period, a decision will be made, ranging from final acceptance to more substantial requests for modification.

You are therefore encouraged, between now and April 21st, to

1. justify your choices (point-by-point response), including those that would not lead to modification, and explain the changes made to the submission (article and code), via the OpenReview platform: you must respond directly to the reviewer, making an “official comment” to each of their reports.
2. In the meantime, update your submission accordingly (HTML, PDF and git repository).

Thank you for submitting to Computo.

---

> ### Author Response · Authors · 2025-04-21
> **Revised Submission Timeline**
>
> Dear Editors and Reviewers,
>
> Thank you for the opportunity to revise and resubmit our manuscript for consideration in Computo. We appreciate the detailed comments provided by the reviewers.
>
> We are currently finalizing our responses to the reviewers’ comments. While we are unable to submit the revised version today, we will upload it by the end of the week. We apologize for the inconvenience and thank you for your understanding.
>
> Best regards,
> Thomas Ferté

---

### Comment · Editors_In_Chief · 2025-05-26
**Production, reproducibility and final publication of your article**

Dear authors,

It is with great pleasure that I confirm the acceptance of your manuscript for publication in Computo.

We currently are  in the production process : to do so, we will clone  your github directory, format your article, associate a DOI, etc. The article and repository will be hosted here: https://github.com/computorg/published-202505-ferte-reservoirnet

Progress will be displayed here : https://github.com/computorg/published-202505-ferte-reservoirnet/issues/1

Any further modification should be made on this repo. We will eventually ask you if you need to make changes, as well as for  proofreading. Changes will be made in the form of a pull-request. We will do our best to guide you through this process.

Thank you for your patience, and thank you for choosing us to publish your research.

---

### Note · Reviewer_2kBF · 2025-04-25

**Comment:**

I would like to thank the authors for their answers and edits, which fully address my comments.

**Audience:**

Yes

**Claims And Evidence:**

Yes

**Decision Recommendation:**

Accept

---

### Note · Reviewer_LM72 · 2025-05-07

**Comment:**

Dear all,

I asked a few questions and suggested some modifications; all of them were addressed, and I am satisfied with the authors' response.

**Audience:**

Yes

**Claims And Evidence:**

Yes

**Decision Recommendation:**

Accept

---

### Note · Action_Editor_bUhc · 2025-05-14

**Comment:**

All reviewers are satisfied that their comments have been taken into account by the authors.

Even if the method's engine, written in Python, has already been enhanced, the reviewers agree on the interest of porting it to R via the wrapper presented here. What's more, the functionalities are presented in abundantly illustrated detail. I recommend acceptance.

**Audience:**

Yes

**Claims And Evidence:**

Yes

**Decision Recommendation:**

Accept

---

### Decision · Action_Editor_bUhc · 2025-05-14

**Recommendation:** Accept as is

**Comment:**

All reviewers are satisfied that their comments have been taken into account by the authors.

Even if the method's engine, written in Python, has already been enhanced, the reviewers agree on the interest of porting it to R via the wrapper presented here. What's more, the functionalities are presented in abundantly illustrated detail. I recommend acceptance.

**Audience:**

Fit to Computo's cope

**Claims And Evidence:**

In agreement with the elements provided by the reviewers, the claims and evidence are sufficiently solid for publication.

---

> ### Decision · Editors_In_Chief · 2025-05-14
>
> I approve the AE's decision.